# CASE-BASED REASONING ENHANCES THE PREDICTIVE POWER OF LLMS IN DRUG-DRUG INTERACTION

## ABSTRACT

Drug–drug interaction (DDI) prediction is critical for treatment safety. While large language models (LLMs) show promise in pharmaceutical tasks, their effectiveness in DDI prediction remains challenging. Inspired by the well-established clinical practice where physicians routinely reference similar historical cases to guide their decisions through case-based reasoning (CBR), we propose CBR-DDI, a novel framework that distills pharmacological patterns from historical cases to improve LLM reasoning for DDI tasks. CBR-DDI constructs a knowledge repository by leveraging LLMs to extract pharmacological insights and graph neural networks (GNNs) to model drug associations. A hybrid retrieval mechanism and two-tier knowledge-enhanced prompting allow LLMs to effectively retrieve and reuse relevant cases, thereby leveraging their in-context learning ability for case-based reasoning. We further introduce a representative sampling strategy for dynamic case refinement. Extensive experiments demonstrate that CBR-DDI achieves state-of-the-art performance, with a significant 28.7% accuracy improvement over both popular LLMs and CBR baseline, while maintaining high interpretability and flexibility.

## 1 INTRODUCTION

Drug-drug interaction (DDI) prediction is critical for pharmacology and healthcare, as it safeguards patients from adverse drug reactions, optimizes therapeutic efficacy, and reduces healthcare costs (Magro et al., 2012; Roemer & Boone, 2013; Marengoni et al., 2014). Accurately identifying DDIs is challenging due to the intricate potential relationships between drugs and the diverse mechanisms underlying the interactions (such as the competition for drug-metabolizing enzymes) (Shen et al., 2024; De Vito et al., 2025). These challenges become even more pronounced when predicting interactions involving new drugs, where interaction data is typically sparse or nonexistent.

Recently, large language models (LLMs) (Brown et al., 2020; Grattafiori et al., 2024; Guo et al., 2025) have demonstrated impressive capabilities across various tasks, particularly excelling at identifying patterns hidden in natural languages. While LLMs have shown promise in pharmaceutical applications (Thirunavukarasu et al., 2023; Liang et al., 2023; Inoue et al., 2024), their effective utilization for DDI prediction remains an open research question. Current approaches commonly enhance LLMs by incorporating biomedical knowledge graphs (KGs) (Xu et al., 2024; Abdullahi et al., 2025), which provide structured knowledge about drugs. They typically employ heuristic methods to retrieve relevant drug information from KGs and feed it directly into LLMs for prediction. However, these methods provide only triplets and are insufficient to activate the reasoning capabilities of LLMs, as drug associations alone cannot reveal their interactions evidently. For example, in Figure 1, the two drugs in the new pair, *Fosphenytoin* and *Diphenhydramine*, both bind to the same gene, yet their actual interaction cannot be directly inferred from this shared property alone. Therefore, modeling and understanding the underlying interaction mechanisms are critical for prediction (De Vito et al., 2025), as they function as intermediate reasoning steps that enhance the LLM's ability to link drug associations with plausible outcomes, analogous to a chain-of-thought process (Wei et al., 2022).

We further observe that many DDI cases share common interaction mechanisms that reflect fundamental pharmacological patterns. As illustrated in Figure 1, the new case and an existing case exhibit similar drug associations, enabling the transfer of known interaction mechanisms from the historical case to the new one. Yet current methods neglect these valuable inter-case relationships, preventing

LLMs from exploiting their powerful in-context learning capacity to perform effective analogy-based reasoning. This also diverges from established clinical practice (Althoff et al., 1998; Bichindaritz & Marling, 2006), where physicians routinely reference historical cases through case-based reasoning (CBR)—a cognitive process that solves new problems by adapting previously solutions to similar problems (Watson & Marir, 1994; Kolodner, 2014).

Inspired by these observations, we propose CBR-DDI, a framework that leverages CBR to enhance LLMs' capabilities for DDI prediction. Our approach constructs a structured knowledge repository that stores a collection of representative cases enriched with pharmacological insights. Each case in the repository includes key associations of drug pair extracted by a GNN module from KGs, and mechanistic insights generated by an LLM, providing a structured representation of pharmacological patterns. To effectively utilize the repository, we design a hybrid retrieval strategy that identifies both semantically and structurally relevant cases, alongside a two-tier knowledge-enhanced prompting to facilitate accurate and faithful reasoning in LLMs. Furthermore, to reduce storage overhead, we propose a sampling strategy that dynamically refines the repository by retaining representative cases. CBR-DDI achieves state-of-the-art performance across multiple benchmarks, outperforming the base LLM model by 463% and surpassing the Naive-CBR baseline by 28.7%. In addition, it offers interpretable prediction results and integrates seamlessly with off-the-shelf

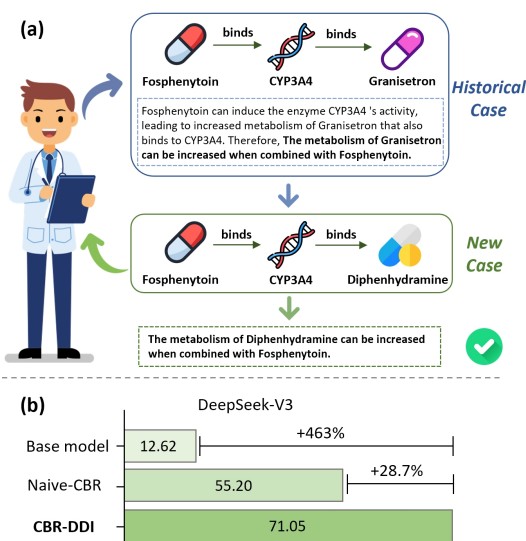

Figure 1: (a). Illustration of using historical cases to solve new cases in DDI task. (b). Accuracy comparison on DrugBank dataset: our CBR-DDI shows significant improvement over base model and Naive-CBR.

LLMs without fine-tuning or intensive interactions. The contributions are summarized as follows:

- Inspired by the success of CBR in clinical practice, we propose CBR-DDI, a new framework that distills pharmacological patterns from historical cases to enhance LLM's reasoning for DDI tasks.

- We propose to construct a knowledge repository, through a collaboration between LLMs for distilling pharmacological insights and GNNs for extracting drug associations from biomedical knowledge graphs.

- For the utilization of the knowledge repository, we design a hybrid retrieval mechanism to identify relevant cases, a two-tier knowledge-enhanced prompting to guide LLMs in case reuse, and a representative sampling strategy for repository refinement.

- Extensive experiments on DDI demonstrate CBR-DDI achieves state-of-the-art performance while maintaining high interpretability and flexibility.

## 2 RELATED WORK

**Drug-Drug Interaction Prediction.** The task of DDI prediction identifies potential adverse interactions or synergistic effects between co-administered medications (Magro et al., 2012). Measuring DDIs in clinical experiments is time-consuming and costly, driving the adoption of machine learning approaches (Luo et al., 2024). *Feature-based methods* leverage shallow models to classify DDI types using drug pair features (e.g., fingerprints) (Rogers & Hahn, 2010; Ryu et al., 2018). *Graph-based methods* model the drug interaction data as a graph. Simple approaches employ embedding techniques (Trouillon et al., 2017; Yao et al., 2022) to learn drug representations. More advanced methods enhance prediction by incorporating biomedical KGs (Himmelstein & Baranzini, 2015; Chandak et al., 2023), which represent relationships between biomedical concepts (e.g., drugs, genes, and diseases) in a multi-relational structure. To capture structural patterns in the graph, various deep models have been proposed, such as graph neural networks (GNNs) (Zitnik et al., 2018; Lin et al., 2020; Yu et al., 2021; Zhang et al., 2023) and graph transformers (Su et al., 2024). *Language model (LM)-based methods* (Zhu et al., 2024) leverage drug descriptions to train models (e.g., RoBERTa

(Liu et al., 2019)) for prediction. Notably, another category of methods (Chen et al., 2021; Zhong et al., 2024; Sun et al., 2025) uses drug molecular structures as input, whereas our approach does not, making these methods orthogonal to ours.

Recently, LLMs are increasingly utilized in biomedical applications, including drug discovery (Chaves et al., 2024), repurposing (Inoue et al., 2024), and molecular understanding (Liang et al., 2023). Their pre-training on vast biomedical literature enables them to leverage implicit knowledge about drug interactions (Sun et al., 2025; De Vito et al., 2025). However, complex drug associations, diverse interaction mechanisms, and multiple interaction types pose significant challenges for LLMs in DDI prediction. Recent approaches heuristically retrieve drug information (e.g., paths between drugs (Abdullahi et al., 2025), one-hop neighbors (Xu et al., 2024)) from KGs and feed it directly into LLMs. However, they fail to explore and leverage the underlying pharmacological mechanisms, reducing the reliability and generalization to new drug prediction.

**Retrieval-Augmented Generation.** Retrieval-Augmented Generation (RAG) (Gao et al., 2023; Huang & Huang, 2024; Baek et al., 2023; Yang et al., 2024) is a framework that enhances the generative capabilities of LLMs by retrieving relevant knowledge from an external knowledge source. Recent advancements have explored to retrieve from KGs to enhance LLMs' reasoning (Pan et al., 2024; Agrawal et al., 2023). These methods primarily extracting question-relevant reasoning paths from KGs for LLMs (LUO et al., 2023; Sun et al., 2023). However, in DDI tasks, explicit questions are absent, and the diverse relational paths between drugs do not directly reveal their interaction type, making these methods difficult to adapt effectively.

**Case-Based Reasoning (CBR).** CBR is a problem-solving paradigm that addresses new problems by adapting solutions from previously resolved cases (Slade, 1991; Watson & Marir, 1994; Kolodner, 2014). Typical CBR process involves retrieving similar past problems, reusing their solutions, evaluating the effectiveness, revising the solution, and retaining successful solutions (Watson & Marir, 1994). Historically, CBR has been widely applied across various domains, such as medical diagnosis (Koton, 1988), and industrial problem-solving (Hennessy & Hinkle, 1992). Recently, there has been increasing interest in integrating CBR with LLMs (Wilkerson & Leake, 2024; Yang, 2024; Guo et al., 2024; Hatalis et al., 2025). However, applying CBR to the DDI task is non-trivial, as it requires carefully designed case retrieval strategies, and existing datasets typically contain only interaction labels without in-depth pharmacological insights as solutions that can be transferred to new cases.

## 3 PROPOSED METHOD

### 3.1 OVERALL FRAMEWORK

In DDI prediction task, we have a set of drugs $\mathcal{V}_\mathcal{D}$ and interaction relations $\mathcal{R}_\mathcal{D}$ among them. Given a query drug pair $(u, v)$, the goal of DDI prediction is to determine their interaction type $r \in \mathcal{R}_\mathcal{D}$. We formulate it as a reasoning task for LLMs to select the most likely interaction type $r$ from the relation set $\mathcal{R}_\mathcal{D}$. Additionally, we utilize a biomedical KG (Himmelstein & Baranzini, 2015) to capture the associations of drugs.

While the diversity of interaction mechanisms presents a significant challenge for DDI prediction, different cases may share interaction patterns, reflecting universal pharmacological principles (Tummino et al., 2021; Roberti et al., 2021). Inspired by the proven success of CBR in clinical practice, we propose CBR-DDI, a framework that distills pharmacological patterns from historical cases to enhance LLM's reasoning. In contrast to naive CBR applications (Brown et al., 2020) that rely on simple retrieval methods (e.g., fingerprint-based matching (Rogers & Hahn, 2010)) and offer only interaction labels as solutions, CBR-DDI constructs a knowledge repository that integrates rich pharmacological insights, and strengthens LLMs through comprehensive case retrieval, knowledge-enhanced reuse, and dynamic refinement.

As illustrated in Figure 2, the framework operates in three stages: (1) case retrieval via LLM-GNN collaboration, (2) case reuse via two-tier knowledge guided reasoning, and (3) case refinement via representative sampling. Given the names of a drug pair, we first leverage the LLM to generate concise drug descriptions, which are used both to perform semantic-level retrieval and to augment a GNN module that encodes the subgraph of the drug pair in the KG. This enables a hybrid retrieval mechanism that identifies both semantically and structurally relevant cases from the knowledge

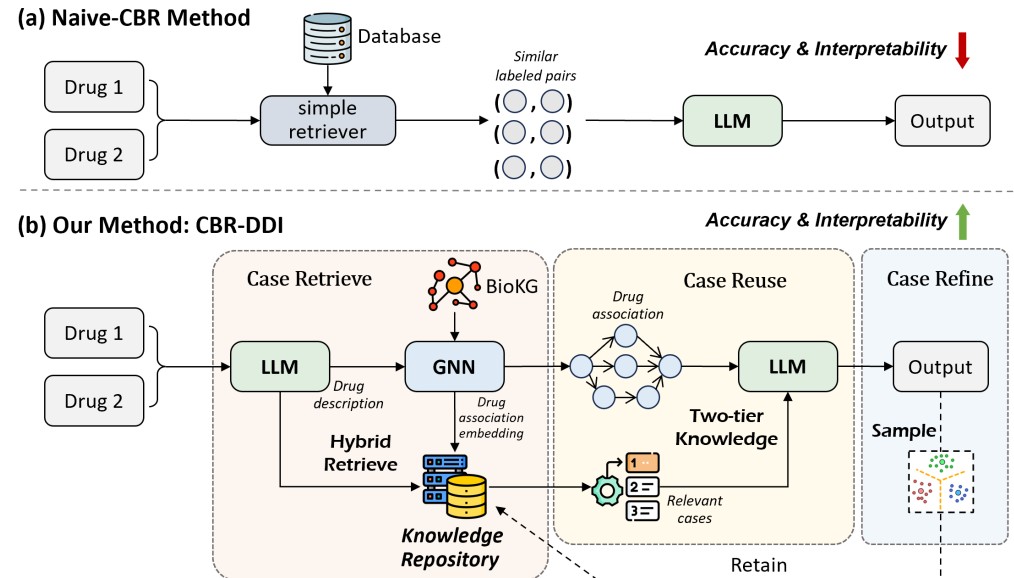

Figure 2: Comparison between Naive-CBR method and our method CBR-DDI. CBR-DDI constructs a knowledge repository storing cases with rich pharmacological insights, and enhances LLM predictions via LLM-GNN collaborative case retrieval, two-tier knowledge-enhanced reuse, and representative sampling-based dynamic refinement.

repository. Then, the retrieved cases are integrated into a two-tier knowledge-enhanced prompt, which combines key drug associations extracted by the GNN module with historically similar mechanistic insights, guiding the LLM to generate accurate and explainable prediction. Finally, we design a sampling strategy to refine the repository by grouping similar cases and retaining representative ones, reducing redundancy and improving adaptability to new discoveries.

## 3.2 KNOWLEDGE REPOSITORY

To effectively leverage the historical drug interaction cases and discover important pharmacological patterns, we propose to construct a lightweight knowledge repository that stores a collection of representative cases enriched with pharmacological insights. This design is inspired by the case-based reasoning paradigm widely adopted in clinical decision support systems (Althoff et al., 1998; Bichindaritz & Marling, 2006), where past cases are enriched and reused to guide new decisions. The repository is designed to

> **<Drug Description>**
> 1. Fosphenytoin: Fosphenytoin is a prodrug of phenytoin, an antiepileptic medication used to treat seizures and other conditions.
> 2. Granisetron: Granisetron is an antiemetic medication used to prevent nausea and vomiting caused by cancer chemotherapy, radiation therapy, and surgery.
> **<Drug Association>**
> (Fosphenytoin, binds, Gene::CYP3A4),(Granisetron, binds, Gene::CYP3A4);
> (Fosphenytoin, resembles, Modafinil),(Modafinil, binds, Gene::CYP3A4), (Granisetron, binds, Gene::CYP3A4) ......
> **<Mechanistic Insights>**
> Fosphenytoin binds to CYP3A4, and Granisetron also binds to CYP3A4. When Fosphenytoin binds to CYP3A4, it can induce the enzyme's activity, leading to increased metabolism of other drugs that also bind to CYP3A4, such as Granisetron. Therefore, The metabolism of Granisetron can be increased when combined with Fosphenytoin.
> **<Interaction Type>**
> The metabolism of Granisetron can be increased when combined with Fosphenytoin.

Figure 3: Illustration of the case.

capture both factual information of drugs and generalizable pharmacological patterns, thereby enabling accurate retrieval of relevant cases and facilitating analogical reasoning in predicting new drug interactions. Specifically, as shown in Figure 3, each case $C$ involving a drug pair $(u, v)$ in the repository is a structured representation of DDIs, consisting of four key components:

- drug description $D_c = (D_u, D_v)$: functional descriptions of the drugs generated by LLM (detailed in Section 3.3.1).;

- drug association $H_c$: structured knowledge extracted from the KG using the GNN module, representing the relationships between drugs, with representation $\boldsymbol{h}_c$ (detailed in Section 3.3.2);

- mechanistic insights $M_c$: pharmacological insights suggest why the drugs might interact, distilled from domain knowledge and historical cases by LLM (detailed in Section 3.3.2);

- interaction type $T_c$: the label of interaction;

Among these, drug descriptions and associations provide factual grounding for retrieval, while the mechanistic insights are the core of each case, as they capture the plausible underlying reason for the interaction and provide key pharmacological patterns that can be transferred to new drug pairs. These patterns guide LLM towards stepwise reasoning process, thereby enhancing its reasoning capacity.

## 3.3 REASONING STEPS

### 3.3.1 CASE RETRIEVAL VIA LLM-GNN COLLABORATION

Effective case retrieval is crucial for CBR, as the relevance and quality of retrieved cases directly impact the accuracy and interpretability of predictions. Considering the diverse functions of drugs and their varying associations, we propose a hybrid retrieval mechanism that combines the natural language processing capabilities of LLMs with the structured learning abilities of GNNs, enabling retrieval of semantically and structurally similar cases.

To retrieve relevant historical cases $C$'s for a given drug pair $p = (u, v)$, we compute a retrieval score based on a weighted combination of semantic similarity and structural similarity:

$$s(p, c) = \lambda \cdot \text{SemanticSim}(p, c) + (1 - \lambda) \cdot \text{StructSim}(p, c), \tag{1}$$

where $\lambda \in [0, 1]$ is a hyperparameter that balances the contribution of the two components. The two similarity are defined as follows:

- SemanticSim$(p, c) = \text{Sim}(f(D_p), f(D_c))$: We prompt an LLM (i.e., Llama3.1-8B-Instruct (Grattafiori et al., 2024)) to generate concise functional descriptions $D_u$ and $D_v$ for drugs $u$ and $v$, denoted as $D_p = (D_u, D_v) = \text{LLM}_{\text{des}}(u, v)$. The function $f(\cdot)$ denotes a text embedding model (Liu et al., 2019). We then compute the cosine similarity between the embeddings of $D_p$ and the stored case description $D_c$, capturing the semantic closeness of drug functionality and pharmacological properties.

- StructSim$(p, c) = \text{Sim}(\boldsymbol{h}_p, \boldsymbol{h}_c)$: We employ a subgraph-based GNN module with attention mechanism (i.e., EmerGNN (Zhang et al., 2023)) to encode the subgraph connecting the drug pair in KG, with the embeddings of LLM-generated drug descriptions as node features, obtaining the subgraph representation: $\boldsymbol{h}_p = \text{GNN}(f(D_u), f(D_v))$. Cosine similarity is then computed between $\boldsymbol{h}_p$ and the stored case representations $\boldsymbol{h}_c$, reflecting the structural similarity in the association patterns between drug pairs.

We rank all cases in the repository based on $s(p, c)$ and select the top-$K$ most relevant ones for subsequent reasoning. By integrating semantic drug descriptions with graph-structured relational knowledge, this hybrid approach enables a comprehensive case retrieval process, capturing pharmacologically similar drug pairs while preserving structural association relevance.

### 3.3.2 CASE REUSE VIA TWO-TIER KNOWLEDGE GUIDED REASONING

Although relevant cases reflect potential interaction mechanisms, they do not provide sufficient factual information for the given drug pair. To address this, we design a two-tier knowledge-enhanced prompt that integrates both external factual knowledge (i.e., drug associations) and internal regularity knowledge (i.e., historical mechanistic insights) to guide the LLM's reasoning process.

Specifically, the prompt comprises the key drug associations of given pair extracted by the attention-based GNN module, and relevant mechanistic insights contained in historical similar cases. The LLM is then prompted to synthesize these two complementary sources of knowledge, generating the interaction mechanism insights $M_p$ and type $T_p$. The prediction process is formalized as:

$$\{M_p, T_p\} = \text{LLM}_{\text{pre}}\big(TD, \{C_i\}_{i=1}^K, H_p, A_p\big), \tag{2}$$

where $TD$ is the task description, $\{C_i\}_{i=1}^K$ are the top-$K$ retrieved cases, $H_p$ denotes the extracted drug association facts, and $A_p$ is the filtered candidate interaction types. We detail the two-tier knowledge as follows:

- External factual knowledge (i.e., drug associations $H_p$): To capture essential associations between drugs, we employ the attention-based GNN module to extract high-quality relational paths that connect them. Unlike prior work (Abdullahi et al., 2025) that retrieves triplets heuristically, we

Table 1: Comparison of different methods using LMs.

| Methods | Without Fine-Tuning | Interpretability | Drug Association Augmentation | Mechanism Augmentation |
|---|---|---|---|---|
| TextDDI | × | × | × | × |
| DDI-GPT | × | ✓ | × | × |
| Naive-CBR | ✓ | × | × | × |
| K-Paths | ✓ | ✓ | ✓ | × |
| CBR-DDI | ✓ | ✓ | ✓ | ✓ |

scores triplets along the paths by attention weights during GNN propagation. We then select the top-$P$ paths with the highest average attention as $H_p$, which are incorporated into the prompt as structured, high-quality factual evidence (e.g., *Fosphenytoin* $\xrightarrow{binds}$ *CYP3A4* $\xrightarrow{binds}$ *Diphenhydramine*).

- Internal regularity knowledge (i.e., mechanistic insights within historical cases $\{C_i\}_{i=1}^{K}$): The retrieved cases (in Section 3.3.1) contain mechanistic insights $M_{c_i}$ that reflect generalized pharmacological patterns observed in similar drug pairs. They can guide the LLM to perform analogical reasoning, drawing parallels between the current drug pair and previously known regularity.

By structuring the prompt in this manner, we enhance the interpretability and reliability of LLM-generated predictions, as the historical cases offer relevant pharmacological patterns, while the factual drug associations provide the evidence base. Furthermore, to reduce the complexity introduced by numerous interaction types, we pre-filter candidate answers $A_p$ based on the scores of GNN module, retaining only top-$N$ candidates. This focuses the LLM's attention on the most plausible options and reduces noise from irrelevant candidates.

### 3.3.3 CASE REFINEMENT VIA REPRESENTATIVE SAMPLING

To ensure both the quality and size control of our knowledge repository, we propose a dynamic refinement strategy that updates cases in the knowledge repository. Specifically, for each LLM-generated prediction, we verify its correctness against ground truth label (i.e., from training data), and prompt revisions for errors based on the correct label. The LLM is also prompted to allow for the possibility of providing no mechanistic insight when neither external knowledge nor its internal knowledge is sufficient; in such cases, we leave this component of the case empty to prevent the introduction of erroneous information. Furthermore, to control the growth of the repository while preserving its expressive power, we group semantically similar cases within each DDI category using the text embeddings of their mechanistic insights $M_c$. Our case-based design allows for simple yet effective clustering methods to retain only the most representative cases, i.e., filtering out redundancy while preserving diversity in pharmacological scenarios (details are shown in Appendix B.1). This approach keeps the repository compact and efficient while allowing for new discoveries.

### 3.4 COMPARISON WITH EXISTING WORKS

As shown in Table 1, TextDDI (Zhu et al., 2024) and DDI-GPT (Xu et al., 2024) rely on fine-tuning small language models (e.g., RoBERTa (Liu et al., 2019)) as classifiers, which limits their compatibility with off-the-shelf LLMs. Specifically, TextDDI relys solely on individual drug descriptions. DDI-GPT retrieves one-hop neighbors from KGs for binary classification and applies an attention mechanism for limited interpretability. Naive-CBR method (Brown et al., 2020) retrieves structurally similar drug pairs based on fingerprint features, providing only case labels for LLMs without deeper pharmacological insight. K-Paths (Abdullahi et al., 2025) uses heuristic methods to extract diverse paths between drugs and directly feeds them into LLMs. In contrast, CBR-DDI uniquely integrates both drug association knowledge and mechanistic insights to augment LLM, enabling interpretable prediction, while offering plug-and-play flexibility across LLMs without requiring fine-tuning.

## 4 EXPERIMENT

### 4.1 EXPERIMENTAL SETUP

**Datasets.** We conduct experiments on two widely used DDI datasets: (1) DrugBank (Wishart et al., 2018), a multi-class dataset that contains 86 types interactions between drugs. (2) TWO-SIDES (Tatonetti et al., 2012), a multi-label dataset that records 200 side effects between drugs.

Table 2: Performance comparison of different methods for DDI. $\Delta_{avg}$ denotes the average improvement in accuracy and recall (in percent) on two datasets.

| Type | Method | DrugBank | | | | TWOSIDES | | | | $\Delta_{avg}$ |
| | | S1 | | S2 | | S1 | | S2 | | |
| | | Acc | F1 | Acc | F1 | Recall | NDCG | Recall | NDCG | |
|---|---|---|---|---|---|---|---|---|---|---|
| Feature-based | MLP | 57.77 | 42.53 | 39.85 | 20.15 | 12.70 | 14.88 | 3.60 | 5.95 | 6.42 ↑ |
| Graph-based | ComplEx | 4.02 | 1.74 | 4.32 | 1.77 | 2.30 | 3.61 | 1.62 | 1.81 | 32.06 ↑ |
| | MSTE | 54.66 | 40.57 | 32.88 | 4.93 | 5.12 | 7.37 | 2.78 | 3.12 | 11.02 ↑ |
| | Decagon | 32.41 | 28.56 | 22.47 | 6.12 | 4.48 | 6.36 | 2.38 | 3.61 | 19.54 ↑ |
| | SumGNN | 57.04 | 54.77 | 25.28 | 17.85 | 4.08 | 5.24 | 2.11 | 3.48 | 13.03 ↑ |
| | EmerGNN | 68.10 | 65.78 | 44.84 | 34.22 | 13.79 | 16.06 | 3.01 | 4.93 | 2.45 ↑ |
| | TIGER | 60.11 | 57.21 | 33.46 | 19.78 | 11.72 | 14.33 | 2.69 | 3.90 | 7.81 ↑ |
| LM-based | TextDDI | 66.75 | 66.53 | 44.23 | 32.79 | 9.88 | 13.24 | 4.16 | 6.04 | 3.35 ↑ |
| Llama3.1-8B | Base | 8.71 | 4.10 | 7.30 | 3.94 | 0.04 | 0.06 | 0.02 | 0.03 | 28.92 ↑ |
| | Naive-CBR | 47.88 | 42.38 | 15.02 | 8.70 | 3.60 | 4.47 | 0.27 | 0.50 | 16.24 ↑ |
| | KAPING | 36.61 | 29.06 | 12.29 | 7.34 | 0.18 | 0.29 | 0.05 | 0.07 | 20.65 ↑ |
| | K-Paths | 57.79 | 46.58 | 35.58 | 22.95 | 0.25 | 0.38 | 0.07 | 0.08 | 9.51 ↑ |
| | CBR-DDI | **68.52** | **61.57** | **44.94** | **32.43** | **13.89** | **15.45** | **4.38** | **7.04** | - |
| Llama3.1-70B | Base | 8.93 | 4.37 | 8.02 | 4.12 | 0.05 | 0.06 | 0.03 | 0.03 | 30.21 ↑ |
| | Naive-CBR | 48.09 | 50.62 | 21.22 | 13.04 | 4.54 | 5.46 | 0.68 | 0.84 | 15.84 ↑ |
| | KAPING | 41.87 | 36.43 | 19.66 | 10.82 | 1.58 | 2.11 | 0.52 | 0.97 | 18.56 ↑ |
| | K-Paths | 61.19 | 56.16 | 36.00 | 24.87 | 2.09 | 3.18 | 1.01 | 1.42 | 9.40 ↑ |
| | CBR-DDI | **71.36** | **70.85** | **47.43** | **36.88** | **14.40** | **16.97** | **4.68** | **7.32** | - |
| DeepSeek-V3 -671B | Base | 12.62 | 9.61 | 12.12 | 6.78 | 0.03 | 0.04 | 0.03 | 0.05 | 28.82 ↑ |
| | Naive-CBR | 55.20 | 47.24 | 22.26 | 15.46 | 3.18 | 4.22 | 0.32 | 0.47 | 14.78 ↑ |
| | KAPING | 46.42 | 40.25 | 22.47 | 14.83 | 1.40 | 1.89 | 0.55 | 0.94 | 17.31 ↑ |
| | K-Paths | 64.52 | 58.17 | 38.33 | 35.41 | 1.73 | 2.21 | 1.19 | 1.66 | 8.58 ↑ |
| | CBR-DDI | **71.05** | **74.38** | **49.45** | **40.69** | **14.85** | **16.56** | **4.73** | **6.60** | - |

Table 3: Comparison of different variants of CBR-DDI-Llama3.1-70B.

| Method | DrugBank | | | | TWOSIDES | | | |
| | S1 | | S2 | | S1 | | S2 | |
| | Acc | F1 | Acc | F1 | Recall | NDCG | Recall | NDCG |
|---|---|---|---|---|---|---|---|---|
| CBR-DDI | 71.36 | 70.85 | 47.43 | 36.88 | 14.40 | 16.97 | 4.68 | 7.32 |
| w.o.case | 68.33 | 68.44 | 46.02 | 33.48 | 13.94 | 15.11 | 3.42 | 5.21 |
| w.o.association | 69.42 | 68.94 | 46.45 | 34.19 | 14.07 | 16.40 | 4.38 | 6.96 |

**Experimental Settings.** Following (Zhang et al., 2023; Abdullahi et al., 2025; Dewulf et al., 2021), we evaluate our model on two challenging settings: S1 and S2. For S1 setting, the task is to predict the interaction type between an new drug—one that has no interaction records in the training set—and an existing drug. For S2 setting, the goal is to predict the interaction type between two new drugs. We also provide experimental results for S0 setting in Appendix C.1.

**Evaluation Metrics.** For DrugBank dataset, where each drug pair corresponds to a single interaction type, we adopt Accuracy and F1 Score as evaluation metrics. For TWOSIDES dataset, where a drug pair may involve multiple interaction types, we treat it as a recommendation task and use Recall@5 and NDCG@5 as the evaluation metrics (detailed demonstrations are provided in Appendix B.3).

**Experiment Details.** We follow the settings of (Zhang et al., 2023) to train the GNN module and use HetioNet (Himmelstein & Baranzini, 2015) as the external KG. Considering the plug-and-play convenience of CBR-DDI, we use three LLMs in experiments: Llama3.1-8B-Instruct (Grattafiori et al., 2024), Llama3.1-70B-Instruct (Grattafiori et al., 2024), and DeepSeek-V3 (Liu et al., 2024). We typically set number of reference cases $K$ as 5, the number of paths in drug associations $P$ as 5, and vary the number of candidate answers among {3,5,10}. Other details are shown in Appendix B.3.

**Baseline Methods.** We consider the following baseline methods for comparison: (1) traditional methods without using LLMs: MLP (Gardner & Dorling, 1998), ComplEx (Trouillon et al., 2017), MSTE (Yao et al., 2022), Decagon (Zitnik et al., 2018), SumGNN (Yu et al., 2021), EmerGNN (Zhang et al., 2023), TIGER (Su et al., 2024), TextDDI (Zhu et al., 2024); (2) LLM-based methods: Base

Table 4: Influence of representative sampling strategy.

| Method | DrugBank | | | | TWOSIDES | | | |
| | S1 | | S2 | | S1 | | S2 | |
| | Acc | #Case | Acc | #Case | Recall | #Case | Recall | #Case |
|---|---|---|---|---|---|---|---|---|
| w.o.sample | 71.36 | 35255 | 47.38 | 3056 | 14.32 | 4684 | 4.68 | 808 |
| w.sample | 71.05 | **2139** | 47.43 | **398** | 14.40 | **1639** | 4.48 | **504** |

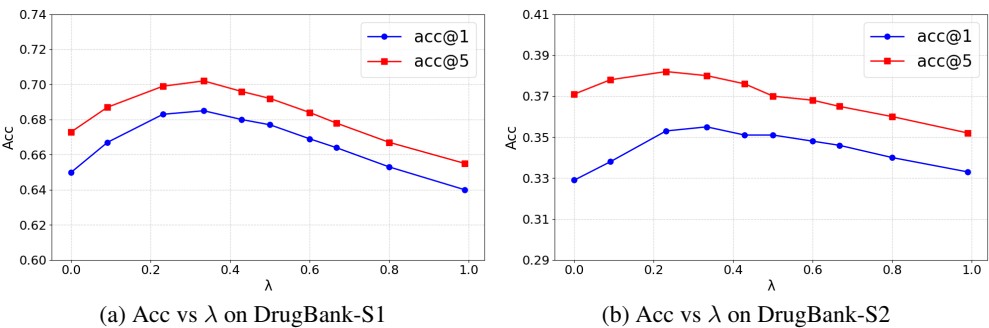

(a) Acc vs $\lambda$ on DrugBank-S1        (b) Acc vs $\lambda$ on DrugBank-S2

Figure 4: Effect of Hyperparameter $\lambda$ on Hybrid Retriever Accuracy.

model, Naive-CBR (retrieve 10 similar labeled cases based on fingerprint similarity as few-shot prompting (Brown et al., 2020)), KAPING (Baek et al., 2023), K-Paths (Abdullahi et al., 2025).

## 4.2 PERFORMANCE COMPARISON

As shown in Table 2, among LLM-based baselines, Naive-CBR achieves notable performance improvements, highlighting the importance of historical cases in prediction. By providing similar drug pairs with their interaction labels, it demonstrates that past interaction patterns offer valuable knowledge for guiding LLM predictions. However, Naive-CBR relies on untrained and simple feature similarity metrics, which fail to capture complex relationships between cases or provide in-depth pharmacological insights. Consequently, it can not outperform other advanced deep learning approaches that are specifically trained for DDI. In contrast, our proposed method, CBR-DDI, significantly outperforms all baseline methods across multiple benchmarks, especially when paired with powerful LLMs like Llama3.1-70B or DeepSeek. Even with smaller models such as Llama3.1-8B, our method achieves superior results over state-of-the-art methods. Compared to heuristic-based approaches like K-Paths, which may introduce irrelevant or redundant information, CBR-DDI effectively leverages historical cases to extract valuable pharmacological insights, and enhances the reasoning capacity of LLMs by integrating both factual drug association knowledge and mechanistic insights, thereby achieving more accurate and reliable predictions. These results demonstrate that CBR-DDI is the first work to effectively unlock the potential of LLMs for DDI prediction.

## 4.3 ABLATION STUDY

### 4.3.1 INFLUENCE OF TWO-TIER KNOWLEDGE AUGMENTATION

To validate the necessity of both factual knowledge (i.e., drug associations) and regularity knowledge (i.e., mechanistic insights derived from historical cases), we conduct ablation studies under three configurations: (i) the full prompt, (ii) factual-only (w.o. case), and (iii) regularity-only (w.o. association). As shown in Table 3, removing any tier of knowledge leads to a performance drop. These results confirm that factual knowledge provides evidence base for reasoning, while regularity knowledge facilitates mechanistic generalization. Notably, the retrieved cases play a more critical role, as drug associations from KGs do not directly determine interaction types. Accurate prediction demands deeper insights into pharmacological mechanisms derived from historical cases, highlighting the importance of case-based reasoning. More detailed ablation studies are presented in Appendix C.2.

### 4.3.2 EFFECTIVENESS OF HYBRID CASE RETRIEVER

We evaluate the effectiveness of the hybrid retriever by varying the similarity weight $\lambda$ between semantic and structural components in (1). Specifically, we measure the retrieval accuracy by

**<Query drug pair-Answer>**
Rifabutin, Zopiclone — The metabolism of Zopiclone can be increased when combined with Rifabutin.
**<Input Task Description>**
You are a medical expert. Your task is to predict the interaction between a pair of drugs. There are some examples for your reference before the given question. You can refer to the interaction mechanisms in the provided examples. You should answer the given question based on the candidate answers, correct probability, related facts and your own knowledge. Please end your reply with `The interaction is <your answer>'.
**<Input Reference Cases>**
**Example:** What is the interaction between *Fosphenytoin and Cytarabine*?
**Related Facts:**
(Fosphenytoin, binds, Gene::CYP2C19), (Glucosamine, binds, Gene::CYP2C19), (Glucosamine, resembles, Cytarabine);
(Fosphenytoin, binds, Gene::CYP3A4), (Cytarabine, binds, Gene::CYP3A4);
(Modafinil, resembles, Fosphenytoin), (The metabolism of Modafinil can be decreased when combined with Glucosamine), (Glucosamine, resembles, Cytarabine);
**Answer:** The given facts suggest that Fosphenytoin binds to CYP2C19 and CYP3A4, and Cytarabine also binds to CYP3A4. Since Glucosamine resembles Cytarabine, it is likely that Cytarabine also binds to CYP2C19. Fosphenytoin's binding to CYP2C19 can lead to increased metabolism of Cytarabine, therefore, The metabolism of Cytarabine can be increased when combined with Fosphenytoin.
**<Input question, drug associations and candidate answers>**
**Question:** What is the interaction between *Rifabutin and Zopiclone*?
**Candidate Answers:**
The serum concentration of Zopiclone can be decreased when it is combined with Rifabutin. (correct probability: 0.251)
The metabolism of Zopiclone can be increased when combined with Rifabutin. (correct probability: 0.233)
The risk or severity of adverse effects can be increased when Rifabutin is combined with Zopiclone. (correct probability: 0.211)
**Related Facts:**
(Rifabutin, resembles, Rifapentine), (Rifapentine, binds, Gene::CYP2C9), (Zopiclone, binds, Gene::CYP2C9);
(Rifabutin, resembles, Rifapentine), (Rifapentine, binds, Gene::CYP3A4), (Zopiclone, binds, Gene::CYP3A4);
(Rifabutin, resembles, Rifapentine),(The metabolism of Eszopiclone can be increased when combined with Rifapentine),(Zopiclone, resembles, Eszopiclone);
**<Output by CBR-DDI-Llama3.1-70B>**
The given facts suggest that Rifabutin resembles Rifapentine and Rifampicin, which increase the metabolism of Eszopiclone, a drug that resembles Zopiclone. This implies that Rifabutin may have a similar effect on Zopiclone, increasing its metabolism. Additionally, Rifapentine binds to CYP2C9, and CYP3A4 enzymes, and Zopiclone also binds to these enzymes. This suggests that Rifabutin may induce these enzymes, leading to increased metabolism of Zopiclone. Considering these facts, the most likely interaction between Rifabutin and Zopiclone is that the metabolism of Zopiclone can be increased when combined with Rifabutin.
The interaction is: The metabolism of Zopiclone can be increased when combined with Rifabutin.

Figure 5: One case study from DrugBank.

selecting the top-$K$ cases ($K = 1, 5$) under different $\lambda$ values and assigning the majority label among them to the test sample. As shown in Figure 4, retrieval accuracy first increases and then decreases as $\lambda$ changes, suggesting that a balanced combination of semantic and structural similarity yields optimal performance. This demonstrates that our hybrid retriever effectively integrates both drug functional descriptions and structural associations, enabling the retrieval of cases that are not only pharmacologically similar but also share interaction patterns, thereby improving prediction accuracy. More experiments are shown in Appendix C.5.

### 4.3.3 INFLUENCE OF REPRESENTATIVE SAMPLING

Table 4 demonstrates the impact of our representative sampling strategy for case refinement. By replacing individual cases with representative cluster centroids, we significantly reduce the size of the case repository—by over 90% in DrugBank—thus greatly enhancing scalability. Notably, reducing the case volume does not compromise performance, while still achieving comparable or even improved results. These results indicate the representative sampling strategy optimizes system efficiency and computational resource usage while filtering out noisy or redundant cases, leading to more representative and informative case selection.

### 4.4 CASE STUDY

We present a case study in Figure 5, which shows the query drug pair, input task description, one of the retrieved cases, extracted drug associations, filtered candidate answers, and the final output of the LLM. As shown, the retrieved case exhibits similar drug associations and mechanisms to those of the query pair, providing strong reasoning evidence. The LLM leverages its powerful in-context learning capabilities to analyze the provided knowledge, generating accurate predictions and mechanistic insights that offer practical value for medical practitioners. This example illustrates how CBR-DDI effectively enhances the LLM's reasoning by incorporating pharmacological insights from historical cases and relevant evidence from KG, resulting in accurate and faithful outcomes.

## 5 CONCLUSION

In this work, we introduce CBR-DDI, a novel framework that leverage CBR to enhance LLMs for DDI tasks. CBR-DDI constructs a knowledge repository by distilling pharmacological insights by an LLM and integrating structured knowledge extracted by a GNN from KGs. The framework employs a hybrid retrieval mechanism for case selection, a two-tier knowledge-enhanced prompting strategy for case reuse, and a representative sampling method for dynamic refinement. Extensive experiments validate the effectiveness of CBR-DDI, achieving state-of-the-art results on multiple benchmarks, while maintaining high interpretability and flexibility.

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

# A DISCUSSION

## A.1 LIMITATIONS AND FUTURE WORK

Like many existing DDI prediction methods leveraging KGs, our approach belongs to the phenotype-based category, where the prediction relies solely on textual information, without incorporating the drug molecular structures. In future work, we believe it is promising to integrate molecular structure processing into our framework, which helps with more precise case retrieval and offer deeper pharmacological insights of interaction mechanism.

In addition, just as the final predictions generated by LLMs may not always be accurate, the mechanistic insights they produce are not guaranteed to be entirely precise. Nevertheless, the interaction labels stored in the knowledge repository can be reliably verified through straightforward string matching with the ground truth provided in the training set. For the mechanistic component, the LLM primarily offers a plausible reasoning path rather than a definitive explanation. To further assess their validity, we conducted additional experiments (Appendix C.2), which demonstrate that these insights are of high quality and play a critical role in supporting the model's reasoning accuracy. Beyond predictive performance, such explicit reasoning traces also enhance interpretability and align with recent findings (Korbak et al., 2025), suggesting that chain-of-thought style outputs can provide new opportunities for monitoring and improving the safety of reasoning models. In future work, it would be valuable to explore how to validate these mechanistic insights by incorporating additional biomedical evidence, or employing advanced verification methods, to further improve the precision of the generated content.

## A.2 REPRODUCIBILITY STATEMENT

Beyond the experimental results presented in the main text, we provide detailed descriptions of the experimental setup in Appendix B.3, including information on datasets, settings, and hyperparameters. In addition, we submit anonymized source code in the supplementary material to further facilitate reproducibility.

## A.3 LLM USAGE

We use LLMs solely as writing assistants to polish the presentation of the paper, such as improving grammar, style, and readability. The LLMs do not contribute to research ideation, methodology or experimental analysis.

## A.4 JUSTIFICATION OF USING LLM TO GENERATE DRUG DESCRIPTIONS

While drug descriptions are largely available in curated databases, the information is often presented in heterogeneous formats and with varying levels of verbosity. An LLM, pre-trained with biomedical knowledge, serves as a powerful tool to synthesize this information into concise and uniform descriptions, which is important for the retrieval and reasoning steps in our pipeline.

Crucially, because this information is grounded in existing databases, we can validate the factual accuracy of the LLM's output. Our validation, we randomly sample 50 drug descriptions generated by Llama3.1-8B-Instruct, and input them with the corresponding ground-truth description from the DrugBank database into a powerful LLM (DeepSeek-R1) to determine whether they are consistent. The evaluation results demonstrate a 98% accuracy rate, confirming that the LLM's output in this step is highly reliable.

# B IMPLEMENTATION DETAILS

## B.1 DETAILS OF KNOWLEDGE REPOSITORY

**Repository Initialization.** To initialize the knowledge repository, we randomly sample a subset of instances from the training data and use them to construct the initial set of cases. For each selected drug pair, we provide the LLM (e.g., Llama3.1-70B-Instruct) with the correct interaction label and

relevant drug association facts, prompting it to generate a plausible mechanistic insight when possible, or to leave this component empty if sufficient knowledge is not available.

**Repository Update.** Whenever the number of cases in the knowledge base exceeds the threshold, or when a certain number of new cases (e.g., 1000) are added, we execute our representative sampling case refinement method. Specifically, we apply the K-Medoids clustering algorithm (Park & Jun, 2009) within each DDI category to group semantically similar cases, using the text embeddings of their mechanistic insights $M_c$. The number of clusters is pre-specified based on the overall sample size (e.g., retaining 5% of the cases or at least 10 cases per category). Within each cluster, only the medoid—the most central and representative case—is retained, while redundant or overly similar cases are removed. This approach not only reduces storage and computational overhead but also ensures that the retained cases reflect diverse pharmacological scenarios.

## B.2 ALGORITHMS FOR GNN MODULE.

Following Zhang et al. (2023), we present the algorithms of the GNN module. Given a drug pair $p = (u, v)$, we implicitly encode the pair-wise subgraph representations with Algorithm 1, and use beam search to find the top-$P$ paths between them with Algorithm 2.

---

**Algorithm 1** Pair-wise subgraph representation learning with flow-based GNN.

---

**Require:** $p = (u, v), \boldsymbol{f}_u = f(D_u), \boldsymbol{f}_v = f(D_v), L, \delta, \sigma, \{\boldsymbol{W}^{(\ell)}, \boldsymbol{w}^{(\ell)}\}_{\ell=1\ldots L}\}, \mathcal{G}.$
  $\{p = (u, v)$: drug pair; $\{\boldsymbol{f}_u, \boldsymbol{f}_v\}$: the embeddings of drug descriptions; $L$: the depth of path-based subgraph; $\delta$: activation function; $\sigma$: sigmoid function; $\{\boldsymbol{W}^{(\ell)}, \boldsymbol{w}^{(\ell)}\}_{\ell=1\ldots L}$: learnable parameters; $\mathcal{G}$: biomedical KG.$\}$
1: initialize the $u \to v$ pair-wise representation as $\boldsymbol{h}_{u,e}^0 = \boldsymbol{f}_u$ if $e = u$, otherwise $\boldsymbol{h}_{u,e}^0 = \boldsymbol{0}$;
2: initialize the $v \to u$ pair-wise representation as $\boldsymbol{h}_{v,e}^0 = \boldsymbol{f}_v$ if $e = v$, otherwise $\boldsymbol{h}_{v,e}^0 = \boldsymbol{0}$;
3: **for** $\ell \leftarrow 1$ to $L$ **do**
4:   **for** $e \in \mathcal{V}_{\mathrm{D}}$ **do** {This loop can work with matrix operations in parallel.}
5:     message for $u \to v$:
$$\boldsymbol{h}_{u,e}^{(\ell)} = \delta\left(\boldsymbol{W}^{(\ell)} \sum_{(e',r,e) \in \mathcal{N}_{\mathrm{D}}} \sigma\left((\boldsymbol{w}_r^{(\ell)})^\top [\boldsymbol{f}_u; \boldsymbol{f}_v]\right) \cdot \left(\boldsymbol{h}_{u,e'}^{(\ell-1)} \odot \boldsymbol{h}_r^{(\ell)}\right)\right);$$
6:     message for $v \to u$:
$$\boldsymbol{h}_{v,e}^{(\ell)} = \delta\left(\boldsymbol{W}^{(\ell)} \sum_{(e',r,e) \in \mathcal{N}_{\mathrm{D}}} \sigma\left((\boldsymbol{w}_r^{(\ell)})^\top [\boldsymbol{f}_u; \boldsymbol{f}_v]\right) \cdot \left(\boldsymbol{h}_{v,e'}^{(\ell-1)} \odot \boldsymbol{h}_r^{(\ell)}\right)\right);$$
7:   **end for**
8: **end for**
9: **Return** $\boldsymbol{h}_p = [\boldsymbol{h}_{u,v}^{(L)}; \boldsymbol{h}_{v,u}^{(L)}]$.

---

**Algorithm 2** Path extractor.

---

**Require:** $(u, v), L, P$
1: initialize openList[0] $\leftarrow u$;
2: set $\mathcal{V}_{u,v}^{(0)} = \{u\}, \mathcal{V}_{u,v}^{(L)} = \{v\}$;
3: obtain the set $\mathcal{V}_{u,v}^{(\ell)} = \{e : d(e, u) = \ell, d(e, v) = L - \ell\}, \ell = 1, \ldots, L$ with bread-first-search;
4: **for** $\ell \leftarrow 1$ to $L$ **do**
5:   set closeList[$\ell$] $\leftarrow \emptyset$, pathList[$\ell$] $\leftarrow \emptyset$;
6:   **for** each edge in $\{(e', r, e) : e' \in \text{openList}[\ell - 1], e \in \mathcal{V}_{u,v}^\ell\}$ **do**
7:     compute the attention weights $\alpha_r^{(\ell)} = \sigma\left((\boldsymbol{w}_r^{(\ell)})^\top [\boldsymbol{f}_u; \boldsymbol{f}_v]\right)$;
8:     compute score($u, e', e$) = score($u, e$) + $\alpha_r^{(\ell)}$;
9:     closeList[$\ell$].add(($e$, score($u, e', e$)));
10:  **end for**
11:  **for** $(u, e', e) \in \text{top}_P(\text{clostList}[\ell])$ **do**
12:    openList[$\ell$].add($e$), pathList[$\ell$].add(($e', r, e$));
13:  **end for**
14: **end for**
15: **Return:** join(pathList[1]...pathList[$L$]).

---

## B.3 DETAILS OF EXPERIMENTS

**Datasets.** We conduct experiments on two widely used DDI datasets: (1) DrugBank (Wishart et al., 2018), a multiclass DDI prediction dataset that contains 86 types of pharmacological interactions between drugs. (2) TWOSIDES (Tatonetti et al., 2012), a multilabel DDI prediction dataset that records 200 side effects between drugs. We use HetioNet (Himmelstein & Baranzini, 2015) as for the external biomedical knowledge graph. Table 5 and 6 display the statistics of the datasets and knowledge graph, where $\mathcal{V}$'s represent the sets of nodes, $\mathcal{R}$'s represent the sets of interaction types, and $\mathcal{N}$'s represent the sets of edges.

Table 5: Statistics of datasets.

| Dataset | $|\mathcal{V}_{\text{D-train}}|$ | $|\mathcal{V}_{\text{D-valid}}|$ | $|\mathcal{V}_{\text{D-test}}|$ | $|\mathcal{R}_{\text{D}}|$ | $|\mathcal{N}_{\text{D-train}}|$ | S1 | | S2 | |
| | | | | | | $|\mathcal{N}_{\text{D-valid}}|$ | $|\mathcal{N}_{\text{D-test}}|$ | $|\mathcal{N}_{\text{D-valid}}|$ | $|\mathcal{N}_{\text{D-test}}|$ |
|---|---|---|---|---|---|---|---|---|---|
| DrugBank | 1,461 | 79 | 161 | 86 | 137,864 | 17,591 | 32,322 | 536 | 1,901 |
| TWOSIDES | 514 | 30 | 60 | 200 | 185,673 | 3,570 | 6,698 | 106 | 355 |

Table 6: Statistics for knowledge graph.

| KG | $|\mathcal{V}_{\text{B}}|$ | $|\mathcal{R}_{\text{B}}|$ | $|\mathcal{N}_{\text{B}}|$ |
|---|---|---|---|
| HetioNet | 34,124 | 23 | 1,690,693 |

**Evaluation metrics.** For the DrugBank dataset, there is one interaction between a pair of drugs. Hence, we evaluate the performance in a multi-class setting, which estimates whether the model can correctly predict the interaction type for a pair of drugs. We consider the following metrics:

- Accuracy: the percentage of correctly predicted interaction type compared with the ground-truth interaction type.
- F1(macro) $= \frac{1}{\|\mathcal{I}_D\|} \sum_{i \in \mathcal{I}_D} \frac{2P_i \cdot R_i}{P_i + R_i}$, where $P_i$ and $R_i$ are the precision and recall for the interaction type $i$, respectively. The macro F1 aggregates the fractions over different interaction types.

**Experimental settings for TWOSIDES.** We modify the traditional setup to accommodate the multi-label nature of TWOSIDES by framing the task as a recommendation problem. The model is required to predict the top 5 most likely side effects for each drug pair, without either binary classification based on predefined interaction type or negative sampling. This design is motivated by three factors: (1) Limitations of conventional classification: binary classification with fixed types and random negatives oversimplifies the task, often producing inflated AUC scores (>90%). (2) Alignment with practical needs: clinicians require identification of potential side effects without prior information, making a multi-label recommendation formulation more realistic. (3) Scalability and LLM efficiency: the traditional setup entails $2 \times 200 = 400$ LLM calls for each drug pair, whereas our approach reduces this to a single call, cutting computational cost by a factor of 400.

Therefore, we use Recall@5 and NDCG@5 as the evaluation metrics:

$$\text{Recall@5} = \frac{|R_{1:5} \cap T|}{|T|}, \tag{3}$$

$$\text{NDCG@5} = \frac{\sum_{i=1}^{5} \mathbb{I}(R_i \in T)^{1}/_{\log_2(i+1)}}{\sum_{i=1}^{\min(|T|,5)} 1/_{\log_2(i+1)}}, \tag{4}$$

where $R$ is a list of recommended interactions for the given pair, $T$ is the ground-truth list, and indicator function $I(x) = 1$ if $x$ is true and 0 otherwise.

**Hyperparameters.** For the training of the GNN module, we follow EmerGNN (Zhang et al., 2023)'s hyperparameter settings. We use three LLMs in experiments: Llama3.1-8B-Instruct (Grattafiori et al., 2024), Llama3.1-70B-Instruct (Grattafiori et al., 2024), and DeepSeek-V3 (Liu et al., 2024). The training of GNN module and the inference of Llama3.1-8B are on an RTX 3090-24GB GPU, while the inference for Llama3.1-70B runs on two A100-80GB GPUs. DeepSeek is accessed via API calls. We set the number of reference cases $K$ to 5, maintain $P = 5$ paths in drug associations, and limit candidate answers to 3 for DrugBank and 10 for TWOSIDES.

Table 7: Performance comparison of different methods for DDI on S0 setting.

| Type | Method | DrugBank | | TWOSIDES | |
|---|---|---|---|---|---|
| | | Acc | F1 | Recall | NDCG |
| Feature-based | MLP | 81.22 | 61.56 | 25.21 | 27.78 |
| Graph-based | Decagon | 87.10 | 58.61 | 12.47 | 14.92 |
| | EmerGNN | 96.48 | 95.44 | 26.84 | 30.22 |
| | TIGER | 95.57 | 93.89 | 21.54 | 25.36 |
| LM-based | TextDDI | 96.04 | 94.53 | 14.07 | 17.64 |
| Llama3.1-70B | Base | 9.17 | 4.79 | 0.06 | 0.07 |
| | Naive-CBR | 57.92 | 54.26 | 7.05 | 8.74 |
| | KAPING | 55.68 | 51.72 | 0.63 | 1.05 |
| | K-Paths | 63.75 | 65.27 | 0.87 | 1.38 |
| | CBR-DDI | **96.98** | **95.95** | **27.18** | **31.04** |

**Baseline Methods.** We consider following baseline methods for performance comparison:

(1) traditional methods without using LLMs:

- MLP (Gardner & Dorling, 1998) uses multilayer perceptron to map the fingerprint features of drugs to the interaction types between them.
- ComplEx (Trouillon et al., 2017) converts KG in to a complex matrix and predict DDI based on the decomposition of the matrix.
- MSTE (Yao et al., 2022) is an embedding-based method that learns on KG to predict the possibility of whether a relation exists.
- Decagon (Zitnik et al., 2018) utilizes drug, genes and diseases information to learn drug representation and predict DDI with a graph convolutional network.
- SumGNN (Yu et al., 2021) samples a subgraph from KG for drug pair and designs a summarization scheme to generate reasoning path in the subgraph.
- EmerGNN (Zhang et al., 2023) designs a flow-based GNN on the KG to learn the representation of subgraph between drugs for prediction.
- TIGER (Su et al., 2024) uses graph transformer to encode the molecular structure and biomedical KG to learn dual-channel representation for drugs.
- TextDDI (Zhu et al., 2024) trains an LM as predictor with an RL-based information selector for extracting relevant drug descriptions.

(2) LLM-based methods:

- Base model is a zero-shot method which directly prompts LLMs to select the most likely interaction type $r$ from the relation set $\mathcal{R}_\mathcal{D}$.
- Naive-CBR (Brown et al., 2020) retrieves 10 similar labeled cases based on fingerprint similarity as few-shot prompting.
- KAPING (Baek et al., 2023) is a universal KG-based RAG approach that retrieves triples via semantic similarity as zero-shot prompting.
- K-Paths (Abdullahi et al., 2025) employs a diversity-aware adaptation of Yen's algorithm to retrieve the K shortest paths between drugs for LLM's prediction.

## C SUPPLEMENTARY EXPERIMENTS

### C.1 PERFORMANCE ON S0 SETTING

We present the performance of different methods under the S0 setting (predicting interactions between existing drugs) in Table 7. As can be seen, our method still achieves the best performance. However, the advantage is not as pronounced as in the S1 and S2 settings, since our approach primarily targets

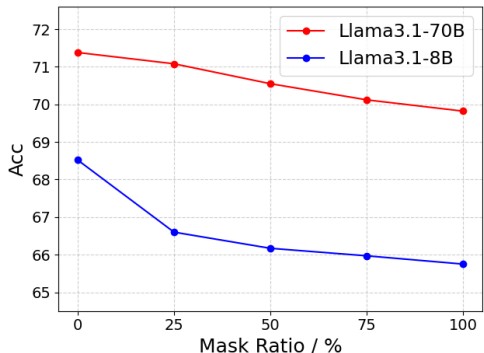

Figure 6: Impact of masking mechanistic insights on DrugBank-S1.

the scenario of new drug prediction. Under the S0 setting, existing methods can memorize possible interaction types between known drugs through training, whereas our method does not fine-tune LLMs and thus lacks this advantage.

## C.2    EFFECT AND RELIABILITY OF MECHANISTIC INSIGHTS

To further evaluate the effectiveness of the mechanistic insights generated by LLMs, we conduct an ablation study on the DrugBank dataset. Specifically, we progressively mask different proportions of the generated insights within each case. The masking ratio, ranging from 0% to 100%, indicates the proportion of the insight content removed (e.g., a ratio of 25% corresponds to removing the last quarter of the insights, whereas 100% corresponds to removing the entire insights, leaving only the interaction label). As illustrated in Figure 6, the performance decreases consistently as the masking ratio increases. These results demonstrate that the LLM-generated mechanistic insights contribute substantially to the prediction process, providing reliable reasoning cues that are indispensable for achieving high accuracy.

In addition, we employ the hallucination detection method proposed in SelfCheckGPT (Manakul et al., 2023) to further assess content validity. Specifically, we randomly sample 500 cases, generate 10 responses for each using the LLM, and computed BERTScore between outputs to estimate the hallucination score. We obtain an average score of $S = 0.21$ (where $S = 0$ indicates no hallucination, S = 1 indicates complete hallucination), suggesting that the LLM outputs are largely consistent and confident with fewer hallucinations. These results provide strong evidence for the effectiveness and reliability of the mechanistic insights in knowledge repository.

## C.3    EFFECT OF CASE NUMBER

We investigate how the number of retrieved cases $K$ affects model performance. As shown in Figure 7, increasing $K$ generally improves accuracy for both the Llama3.1-8B and 70B models. These results suggest that incorporating more cases enhances LLM's reasoning by providing richer phamacological insights, but overly large $K$ may introduce redundancy or noise. Specifically, incorporating case information can significantly enhance the performance of smaller LLMs (i.e., Llama3.1-8B), as their weaker reasoning capabilities make it difficult to delve beyond superficial drug associations to uncover underlying interaction mechanisms and consequently make accurate predictions.

## C.4    EFFECT OF DRUG ASSOCIATION KNOWLEDGE

We also analyze the impact of the number of extracted drug association paths $P$ on model performance. As shown in Figure 8, prediction accuracy initially improves with increasing $P$, as additional paths provide more factual evidence for mechanistic reasoning. However, beyond an optimal point, performance gradually declines as excessive paths introduce irrelevant or conflicting relationships that obscure core interaction mechanisms.

Furthermore, the Figure compares our attention-based GNN retriever with the random retriever (i.e., heuristic retrieval used in existing methods). The results demonstrate that our GNN retriever

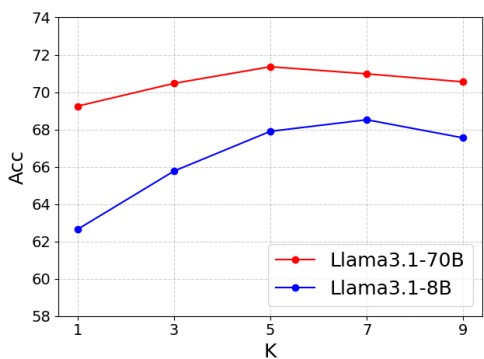

Figure 7: Impact of the number of retrieved cases on DrugBank-S1.

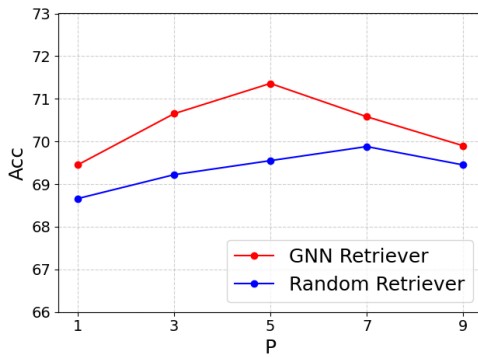

Figure 8: Impact of retrieved drug associations on DrugBank-S1 of CBR-DDI-Llama3.1-70B.

achieves superior performance, as the attention mechanism enables the model to learn and prioritize more high-quality relationship paths, thereby providing a more effective foundation for reasoning. In contrast, heuristic retrieval methods lack this discriminative capability to identify the critical pharmacological relationships.

## C.5 EFFECT OF HYBRID RETRIEVER

In Figure 4, we have conducted an ablation study on different retrieval strategies, in terms of the retrieval accuracy. In that figure, the leftmost point ($\lambda = 0$) corresponds to using only structural similarity (StructSim), while the rightmost point ($\lambda = 1$) corresponds to using only semantic similarity (SemanticSim). The results indicate that a balanced combination of semantic and structural similarity achieves the best retrieval performance.

To further substantiate this finding, we additionally evaluate the impact of each retriever on the final prediction accuracy of the LLM on DrugBank-S1 dataset. As reported in Tabel 8, the hybrid retriever outperforms both individual components, which aligns with the trends observed in Figure 4.

Table 8: Performance of different retrievers on DrugBank-S1.

| Retriever | SemanticSim | StructSim | Hybrid |
|-----------|-------------|-----------|--------|
| Accuracy  | 69.43       | 70.74     | 71.36  |

We also present the most relevant cases retrieved by different retrievers for the same query drug pair (Figure 9). When using either the semantic-based retriever ($\lambda = 1$) or the structure-based retriever ($\lambda = 0$) alone, the retrieved cases often fail to share the same interaction type as the test case and therefore provide limited value for mechanism reasoning. In contrast, the proposed hybrid retriever effectively combines semantic similarity and structural similarity, enabling the retrieval of cases that

capture both relevant pharmacological effects and meaningful drug associations. For clarity, we omit the mechanistic insights in the retrieved cases, since they are not involved in the retrieval process.

## C.6  ADDITIONAL PERFORMANCE COMPARISON

Table 9: Performance Comparison on TWOSIDES dataset.

| Method | S1-Recall | S1-NDCG | S2-Recall | S2-NDCG |
|--------|-----------|---------|-----------|---------|
| SA-DDI | 4.43 | 5.98 | 3.98 | 5.96 |
| CBR-DDI | **14.40** | **16.97** | **4.68** | **7.32** |

As noted in the related work section, several DDI methods rely on molecular structure graphs, whereas our method does not incorporate such structural information, making it fundamentally orthogonal to these approaches. For this reason, they are not included in the main experimental results.

To provide a more comprehensive comparison, we additionally evaluate a strong baseline SA-DDI (Yang et al., 2022) among these methods. Specifically, for each query drug pair, SA-DDI is used to compute scores for all possible side effects, and the top-$K$ side effects ($K = 5$) were taken as its final predictions. The results are displayed in Table 9, which show that our method consistently outperforms SA-DDI across all metrics on the TWOSIDES dataset.

# D  CASE STUDY

We present two more representative cases from DrugBank and TWOSIDES in Figure 10 and Figure 11. Each case includes LLM-generated drug descriptions, key drug associations extracted by the GNN module, mechanistic insights generated by the LLM based on both external and internal knowledge, and the ground truth interaction label. These cases are constructed to capture both factual evidence and underlying pharmacological patterns of drug interactions, thereby supporting accurate retrieval and interpretable reasoning for new prediction tasks.

--------------------------------------------------------- *Query Drug Pair* ---------------------------------------------------------

*<Drug Description>*
1. Maprotiline: Maprotiline is a tetracyclic antidepressant (TeCA) used to treat major depressive disorder. It works by increasing the levels of certain neurotransmitters in the brain …….
2. Cinacalcet: Cinacalcet is a calcimimetic agent used to treat secondary hyperparathyroidism in patients with chronic kidney disease on dialysis, as well as to treat hypercalcemia in patients with parathyroid carcinoma.

*<Drug Association>*
(Maprotiline, binds, Gene::CYP2D6), (Cinacalcet, binds, Gene::CYP2D6);
(Maprotiline, binds, Gene::CYP1A2), (Cinacalcet, binds, Gene::CYP1A2);
(Maprotiline, binds, Gene::CYP3A4), (Cinacalcet, binds, Gene::CYP3A4) ……

*<Interaction Type>*
The metabolism of Cinacalcet can be decreased when combined with Maprotiline.

-------------------------------------------- *Retrieved Case with Hybrid Retriever* ----------------------------------------------

*<Drug Description>*
1. Maprotiline
2. Cimetidine: Cimetidine is a histamine H2-receptor antagonist that is used to treat ulcers and gastroesophageal reflux disease (GERD) by reducing stomach acid production.

*<Drug Association>*
(Maprotiline, binds, Gene::CYP2D6), (Cimetidine, binds, Gene::CYP2D6);
(Maprotiline, binds, Gene::CYP1A2), (Cimetidine, binds, Gene::CYP1A2);
(Maprotiline, binds, Gene::ABCB1), (Cimetidine, binds, Gene::ABCB1)……

*<Interaction Type>*
The metabolism of Cimetidine can be decreased when combined with Maprotiline. ✅

------------------------------------------ *Retrieved Case with Semantic Retriever* -------------------------------------------

*<Drug Description>*
1. Maprotiline
2. Pomalidomide: Pomalidomide is an immunomodulatory drug used in the treatment of multiple myeloma, a type of blood cancer. It works by inhibiting the growth of cancer cells and enhancing the immune system's ability to attack cancer cells.

*<Drug Association>*
(Maprotiline, binds, Gene::ABCB1), (Pomalidomide, binds, Gene::ABCB1);
(Maprotiline, binds, Gene::CYP1A2), (Pomalidomide, binds, Gene::CYP1A2);
(Maprotiline, resemble, Desiparamine), (Desipramine, binds, Gene::ABCB1), (Pomalidomide, binds, Gene::ABCB1) ……

*<Interaction Type>*
The risk or severity of adverse effects can be increased when Maprotiline is combined with Pomalidomide. ❌

------------------------------------------ *Retrieved Case with Structure Retriever* -------------------------------------------

*<Drug Description>*
1. Maprotiline
2. Desvenlafaxine: Desvenlafaxine is a serotonin-norepinephrine reuptake inhibitor (SNRI) used to treat major depressive disorder and generalized anxiety disorder.

*<Drug Association>*
(Maprotiline, binds, Gene::CYP3A4), (Desvenlafaxine, binds, Gene::CYP3A4);
(Maprotiline, binds, Gene::CYP2D6), (Desvenlafaxine, binds, Gene::CYP2D6);
(Maprotiline, binds, Gene::SLC6A2), (Desvenlafaxine, binds, Gene::SLC6A2) ……

*<Interaction Type>*
The risk or severity of adverse effects can be increased when Maprotiline is combined with Desvenlafaxine. ❌

Figure 9: Retrieved cases of different retrievers on DrugBank-S1.

*<Drug Description>*
1. Betaxolol: Betaxolol is a beta-blocker medication used to treat high blood pressure and glaucoma.
2. Salmeterol: Salmeterol is a long-acting beta-2 adrenergic receptor agonist (LABA) used to treat asthma and chronic obstructive pulmonary disease (COPD).

*<Drug Association>*
(Betaxolol, binds, Gene::ADRB2),(Salmeterol, binds, Gene::ADRB2);
(Betaxolol, binds, Gene::ADRB1), (Propafenone, binds, Gene::ADRB1),(Propafenone, resembles, Salmeterol);
(Betaxolol, binds, Gene::ADRB1), (Arbutamine, binds, Gene::ADRB1), (Arbutamine, resembles, Salmeterol);
(Betaxolol, binds, Gene::ADRB2), (Salbutamol, binds, Gene::ADRB2), (Salbutamol, resembles, Salmeterol);
(Betaxolol, binds, Gene::CYP2D6),(Labetalol, binds, Gene::CYP2D6),(Salmeterol, resembles, Labetalol).

*<Mechanistic Insights>*
The given facts suggest that Betaxolol binds to both ADRB1 and ADRB2 receptors. Salmeterol also binds to ADRB2 receptors, which are responsible for bronchodilation. Since Betaxolol binds to ADRB1 and ADRB2 receptors, it can potentially block the action of Salmeterol on ADRB2 receptors, leading to decreased bronchodilatory activities of Salmeterol. Therefore, Betaxolol may decrease the bronchodilatory activities of Salmeterol.

*<Interaction Type>*
Betaxolol may decrease the bronchodilatory activities of Salmeterol.

Figure 10: One case from DrugBank.

*<Drug Description>*

1. Loratadine: Loratadine is a non-sedating antihistamine used to treat allergies, such as hay fever, and skin conditions like hives.
2. Nifedipine: Nifedipine is a calcium channel blocker used to treat high blood pressure and angina (chest pain).

*<Drug Association>*

(Loratadine, resemmbles, Azatadine), (Azatadine, resembles, Cyproheptadine),(Cyproheptadine, upregulates, Gene::ADO),(Nifedipine, upregulates, Gene::ADO);

(Loratadine, resemmbles, Azatadine), (Azatadine, resembles, Cyproheptadine),(Cyproheptadine, upregulates, Gene::RRP8),(Nifedipine, upregulates, Gene::RRP8);

(Loratadine, resemmbles, Azatadine), (Azatadine, resembles, Cyproheptadine),(Cyproheptadine, upregulates, Gene::RRS1),(Nifedipine, upregulates, Gene::RRS1)

*<Mechanistic Insights>*

The given facts suggest that Loratadine and Nifedipine may interact through their effects on certain genes, including Gene::ADO, Gene::RRP8, and Gene::RRS1. While the exact mechanisms are unclear, we can make some educated guesses about the potential side effects.  The upregulation of these genes by both Loratadine (via its resemblance to Cyproheptadine) and Nifedipine may lead to increased expression of certain proteins involved in inflammation and vascular regulation. This could contribute to the development of acne rosacea, a condition characterized by inflammation and increased blood flow to the skin.  The interaction between Loratadine and Nifedipine may also affect blood pressure regulation, leading to increased pressure in the portal vein, which could cause portal hypertension.  Finally, the upregulation of Gene::ADO, Gene::RRP8, and Gene::RRS1 may also affect the urinary system, leading to changes in urine production and composition, which could result in pyuria (the presence of pus in the urine).  Therefore, when Loratadine and Nifedipine are used together, the side effects are [acne rosacea, portal hypertension, pyuria].

*<Interaction Type (Side Effects)>*

[acne rosacea, portal hypertension, pyuria]

Figure 11: One case from TWOSIDES.

