# OpenReview forum: "Case-Based Reasoning Enhances the Predictive Power of LLMs in Drug-Drug Interaction"
_ICLR.cc/2026/Conference — Submitted to ICLR 2026_

### Official Review · Reviewer_kDBh · 2025-10-27

**Soundness:** 2
**Presentation:** 2
**Contribution:** 2
**Rating:** 4
**Confidence:** 4

**Summary:**

This paper proposes CBR-DDI, a framework that distills pharmacological patterns from historical cases to improve LLM reasoning for DDI (Drug-Drug Interaction) tasks.

**Strengths:**

Applying LLMs for drug-drug interaction prediction is interesting

**Weaknesses:**

- Data Leakage Handling: How was data leakage addressed? While the evaluation protocol confirms evaluation on new drugs, there is no guarantee that the LLM has not been trained on these new drugs. The fact that the LLM uses the drug name instead of the SMILES (Simplified Molecular-Input Line-Entry System) for generating descriptions suggests a possibility that the LLM may already possess information about the drug based on its name.

- Knowledge Graph Coverage: Is it possible for test drugs to be absent from the knowledge graph? If so, how are these cases handled?

- Relationship between Retrieval Accuracy and Performance (Experiment 4.3.2): What is the relationship between retrieval accuracy and the actual performance in Experiment 4.3.2? Does accurate retrieval lead to an increase in actual performance?

**Questions:**

See Weaknesses Section

---

> ### Author Response · Authors · 2025-11-21
> **Response to Reviewer kDBh**
>
> Thank you for your valuable feedback and insightful questions. We appreciate the opportunity to clarify these important aspects of our work regarding potential data leakage, knowledge graph coverage, and the retrieval process.
>
> **W1：About Data Leakage.**
>
> **A1:** We agree that data leakage is a critical concern when working with large pre-trained models. We have carefully considered this and believe our evaluation protocol is robust against this issue.
>
> While the LLM was pre-trained on a vast corpus that likely includes medical literature, and therefore may possess basic information about a drug from its name, **this knowledge does not extend to drug-drug interaction outcomes**. Our experimental results strongly support this claim:
>
> - The **Base LLM models perform extremely poorly** when prompted directly to predict DDIs. As shown in Table 2, the powerful DeepSeek-V3 (671B) model achieves an accuracy of only **12.6%** on the DrugBank dataset. This is significantly lower than even simple, traditional machine learning baselines like MLP (57.8%) and demonstrates that the LLM cannot recall DDI outcomes from its pre-training data.
> - In contrast, when augmented with our **CBR-DDI **framework, the same LLM's accuracy jumps to **71.05%**. This dramatic improvement, which far surpasses other LLM-based methods, shows that our framework's success comes from its ability to effectively harness the LLM's reasoning capabilities through CBR, not from exploiting memorized DDI knowledge.
>
> Furthermore, we focus our main experiments on the "new drug" settings, because they represent a more challenging scenario. We also conducted experiments in the **S0 setting** (predicting drugs with some known interactions), with results presented in **Appendix C.1**. These results also confirm the superior performance of our method, further validating its effectiveness.
>
>
>
> **W2：About Knowledge Graph Coverage.**
>
> **A2:** In our experiments, we follow a standard and widely adopted setting from existing literature. In this setup, a "new drug" does not appear in the training set of the DDI network (which only contains drugs and their interaction labels), but it **is present in the broader biomedical knowledge graph (KG)**. This is a reasonable assumption that reflects a real-world scenario: when a new drug is developed, it doesn't have a history of observed interactions with other drugs, but its basic pharmacological properties—such as its known targets (genes, proteins) or its therapeutic class (diseases)—are documented and can be represented in a KG.
>
> However, in the hypothetical case where a drug is entirely absent from the KG, our CBR-DDI framework is designed to handle this gracefully. The structural similarity component of our hybrid retriever would be omitted for that drug. The retrieval of historical cases would then rely solely on the **semantic similarity** of the drug descriptions. The rest of the reasoning pipeline would proceed as usual, allowing the model to make a prediction based on the best available semantic information.
>
> **W3：About Relationship between Retrieval Accuracy and Performance.**
>
> **A3:** There is indeed a direct and positive relationship: **more accurate case retrieval leads to better final prediction performance.**
>
> The experiment described in Section 4.3.2 primarily evaluates the *retrieval accuracy* itself by varying the hyperparameter λ. To explicitly connect this to the final prediction performance, we conducted an additional ablation study presented in **Appendix C.5 and Table 8** (also shown below). This study evaluates the final performance of the CBR-DDI using different retrievers. The results idemonstrate that the **hybrid retriever outperforms both individual components**, leading to the highest final prediction accuracy. This finding directly aligns with the retrieval accuracy trends observed in Figure 4 and confirms that the improved retrieval quality of our hybrid approach is a key driver of the framework's overall superior performance.
>
> **Table 8**: Performance of different retrievers on DrugBank-S1.
>
> | Retriever | SemanticSim | StructSim | Hybrid |
> | :-------: | :---------: | :-------: | :----: |
> | Accuracy  |    69.43    |   70.74   | 71.36  |
>
>
> We hope these clarifications have adequately addressed your questions. Thank you again for your thoughtful review.

---

> > ### Comment · Reviewer_kDBh · 2025-11-25
> >
> > Thank you for your efforts during rebuttal.
> >
> > However, I still not sure about extrapolating capability of the model. As far as I know, biomedical KG already contains the drug-drug interaction scenarios, and the model should be experimented on the case where the drugs are not appeared in any KGs.

---

> > > ### Author Response · Authors · 2025-11-26
> > >
> > > Thank you for your feedback.
> > >
> > > We believe there may be a misunderstanding about what information the biomedical KG contains, as the **KG does not contain the drug-drug interaction labels** that we are aiming to predict. Our experimental setup involves two distinct and separate data sources:
> > >
> > > 1. **DDI Network** (e.g., DrugBank, TWOSIDES): This dataset provides our ground-truth labels. It consists of pairs of drugs and their known interaction types, such as (*DrugA, increases_metabolism_of, DrugB*)).
> > > 2. **Biomedical Knowledge Graph** (e.g., HetioNet): This is an external knowledge source. It contains relationships between biomedical concepts (e.g., drugs, genes, and diseases),  such as (*Drug, binds_to, Gene*), (*Drug, treats, Disease*). **It does not contain the DDI relationships.**
> > >
> > > As a standard and widely adopted setting, the model's task is to **learn how to reason over the factual knowledge in the KG** to **predict the interaction labels** from the DDI dataset. Therefore, the KG provides the factual evidence, not the answers.
> > >
> > > This distinction is key to understanding the model's extrapolative capability. A "new drug" is one that is absent from our DDI training set, but its fundamental properties (its links to genes, proteins, etc.) are present in the KG. The model must extrapolate from known DDI cases and the new drug's connections within the KG to infer a completely new interaction.
> > >
> > > We hope this clarifies the distinction between our label source and our knowledge source. The model is indeed extrapolating, as it must learn a reasoning process over fundamental biology rather than memorizing interactions. We will ensure this distinction is made crystal clear in the revised manuscript.

---

> ### Author Response · Authors · 2025-11-25
>
> Dear Reviewer,
>
> We are writing to follow up on your review for our paper. First, we'd like to thank you again for the time you took to provide such detailed and constructive feedback.
>
> We have posted an official comment addressing the concerns you raised, including potential data leakage, knowledge graph coverage, and the retrieval impact.
>
> We would be very grateful if you had a moment to consider our response. We hope it helps address the issues you identified.
> Thank you once again for your valuable engagement with our work.
>
> Sincerely,
>
> Authors

---

### Official Review · Reviewer_6rWq · 2025-10-30

**Soundness:** 2
**Presentation:** 3
**Contribution:** 3
**Rating:** 4
**Confidence:** 3

**Summary:**

This paper presents CBR-DDI, an LLM-based method for predicting drug-drug interactions (DDIs). CBR-DDI uses a combination of an LLM and a GNN to incorporate drug interaction data with historical context about its use. By doing this, CBR-DDI better reflects the clinical practice of using previous cases to inform current ones. Overall, the authors have completed lots of interesting work to showcase the approach, comparing it against a suite of baseline methods and providing a case study of its success. However, it was unclear why an LLM was used at the beginning of the pipeline (and whether this was an appropriate choice) and why the authors tested their approach on new drugs. Please see below for the clarifications I would suggest.

**Strengths:**

**Strong points:**
- The authors benchmarked the approach against a suite of many diverse approaches. I really applaud the authors for choosing benchmarks from both graph-based and LM-based approaches.
- Section 3.3.1: The approach here which balances semantic and structural similarity is very clever and sensible with respect to the task at hand.
- Well done to the authors for adjusting the metrics appropriately based on the nature of each dataset (DrugBank and TWOSIDES)- it shows that the authors really familiarized their selves with the datasets.
- For the most part, the claims made follow from the results provided.
- DDI prediction is much less researched than other drug discovery tasks, so it is an important area of research.
- Really nice figures.

**Weaknesses:**

**Weak points**

- Sections 3.2 and 3.3.2: Why use an LLM to generate drug descriptions or mechanism insights? Such descriptions are available on professionally curated databases (e.g., https://go.drugbank.com/drugs/DB00945 or https://pubchem.ncbi.nlm.nih.gov/compound/Aspirin). LLMs are infamous for hallucinations and mistakes in scientific disciplines. Thus, I would argue that this step, alone, needs to be validated before proceeding with the whole pipeline. For example, the authors ask the LLM to generate mechanistic insights, but I would suggest that you need to, somehow, verify that it can reliably generate those. Perhaps https://github.com/SuLab/DrugMechDB could help.
- It is not entirely clear why the authors tested the approach on drug pairs between new and existing or both new drugs. In the beginning, the authors even say: “These challenges become even more pronounced when predicting interactions involving new drugs, where interaction data is typically sparse or nonexistent.” If historical context is unavailable for new drugs, then how is this a useful approach for such drug pairs?
- A couple sentences in the introduction are difficult to understand:
    - “However, these methods provide only triplets and are insufficient to activate the reasoning capabilities of LLMs, as surface-level drug associations alone cannot reveal their potential interactions evidently” -->
        - The terms “surface-level” and “potential interactions” are quite vague so it’s not clear what’s meant here.
        - It’s also not clear what it means to “activate the reasoning capabilities of LLMs”.
        - Why are triplets not sufficient? Is there a paper that demonstrates this insufficiency?
    - “For example, in Figure 1, the new drug pair Fosphenytoin-Diphenhydramine binds to the same gene, yet their actual interaction cannot be directly inferred.”
        - The way this sentence is structured is confusing. The subject of the sentence is singular, but then "their" is used. Also, the way it's worded implies that a "drug pair" is one entity which binds to a gene.
        - This sentence also seems to imply that interactions between drugs are literal, physical interactions between drugs as opposed to interferences or cumulative effects on biological systems.

**Questions:**

I would ask the authors to provide some clarification or validation with respect to the above weak points.

1. Can the authors please justify why the use of an LLM to generate drug functional descriptions and interaction mechanisms, as opposed to using curated resources available from databases?
2. Can the authors either validate, experimentally, that the output of the first LLM step is reliable or correct, or justify why such validation is not necessary?
3. Please elaborate on why testing was done on new drugs and how this fits into the use of historical context and CBR.
4. Could the authors please clarify the above sentences in the Weaknesses section?

---

> ### Author Response · Authors · 2025-11-21
> **Response to Reviewer 6rWq (Part 1）**
>
> Thank you for your detailed feedback and insightful questions. We appreciate the opportunity to clarify these important aspects of our work.
>
> **Q1: About the use of an LLM to generate drug functional descriptions and interaction mechanisms.**
>
> **A1:**
> - **For Drug Descriptions:** Our primary goal was **synthesis and standardization**. Information across different databases is often heterogeneous in format and verbosity. An LLM, pre-trained with biomedical knowledge, can synthesize this information into concise and uniform descriptions. This standardized format is crucial for effective retrieval and reasoning in the subsequent steps of our pipeline. Moreover, we validate the correctness of the generated drug description in **A2** below.
> - **For Interaction Mechanisms:** Mechanistic explanations for drug-drug interactions are **largely absent from existing databases**. While the GitHub repository you suggested is a great resource, it primarily documents drug-disease relationships, whereas our focus is on drug-drug interactions. In addition, manually curating this knowledge is labor-intensive and slow. Our work proposes a method to automate the generation of the possible mechanisms, offering a more efficient and scalable approach that is vital for long-term development in this field.
>
>
>
>
> **Q2: About the validating of the output of first LLM step.**
>
> **A2：** We agree that validating the LLM's output is crucial. To address this, we conducted an additional experiment.
>
> We randomly sampled 50 drug descriptions generated by Llama3.1-8B-Instruct. To evaluate their factual accuracy, we input both the drug description generated by LLM and the corresponding ground-truth description from the DrugBank database into a powerful LLM (DeepSeek-R1) to determine whether they are consistent. **The evaluation showed a 98% accuracy rate**, confirming that the LLM's output is highly reliable. This is because the  LLMs have been trained on biomedical corpora, endowing them with basic information of drugs.
>
> The single failure case was for the drug "Epirizole", for which the LLM stated, "I couldn't find any information on a medication called Epirizole." An inspection of the DrugBank database reveals that this drug's attributes for 'US Approved' and 'Other Approved' are both marked 'NO'. This indicates the drug has not been approved for market, which explains why it would fall outside the scope of the LLM's knowledge.
>
> We have provided an example demonstrating the reliability of the LLM-generated content, where the drug description generated by the LLM is not only more comprehensive than the DrugBank summary, but also more concise than the detailed drug background:
>
> - **LLM-Generated Description**: "Quinidine is an antiarrhythmic medication used to treat abnormal heart rhythms, such as atrial fibrillation and ventricular tachycardia. It works by blocking certain electrical signals in the heart, helping to restore a normal heartbeat."
> - **Drug Summary on DrugBank**: "Quinidine is a medication used to restore normal sinus rhythm, treat atrial fibrillation and flutter, and treat ventricular arrhythmias."
> - **Drug Background on DrugBank**: "Quinidine is a D-isomer of quinine present in the bark of the Cinchona tree and similar plant species. This alkaloid was first described in 1848 and has a long history as an antiarrhythmic medication. Quinidine is considered the first antiarrhythmic drug (class Ia) and is moderately efficacious in the acute conversion of atrial fibrillation to normal sinus rhythm. It prolongs cellular action potential by blocking sodium and potassium currents. A phenomenon known as “quinidine syncope” was first described in the 1950s, characterized by syncopal attacks and ventricular fibrillation in patients treated with this drug. Due to its side effects and increased risk of mortality, the use of quinidine was reduced over the next few decades. However, it continues to be used in the treatment of Brugada syndrome, short QT syndrome and idiopathic ventricular fibrillation."
>
> We will add a new section to the appendix detailing this validation study to further strengthen the paper. Thank you for this valuable suggestion.

---

> ### Author Response · Authors · 2025-11-21
> **Response to Reviewer 6rWq (Part 2）**
>
> **Q3: About the test setting of new drugs.**
>
> **A3:** Our framework is effective in both settings that include new drugs and those that do not. As shown in **Appendix C.1**, our method also achieves SOTA performance in the S0 setting (interactions between known drugs). We choose to highlight the S1 and S2 settings in the main paper because they are widely recognized as more challenging and meaningful scenarios.
>
> A **"new drug"** in this context refers to a drug with **no known DDI labels in the training set**, but for which fundamental properties (e.g., its targets, pathways) are known and **exist in the biomedical KG**. This mirrors the real-world scenario of developing a new drug, where its basic functions are understood before all its interactions are tested.
>
> This setup is precisely where our CBR approach excels. Since the new drug has known properties and connections in the KG, our hybrid retriever can identify pharmacologically similar drugs from the knowledge repository. The model then uses the reasoning patterns from these analogous past cases to infer the interaction for the new drug, directly addressing the critical cold-start problem.
>
>
>
> **Q4: About two controversial sentences.**
>
> **A4:** Thank you for pointing out these ambiguities. We will revise them for clarity.
>
> - **Regarding the sentence: "However, these methods provide only triplets..."**
>   - **"Surface-level"** refers to factual triplets like (Fosphenytoin, binds_to, CYP3A4). These state a static association but do not capture the functional consequence (e.g., does it induce or inhibit the enzyme?). The **"potential interactions"** are the true interaction types we aim to predict.
>   - **"Activate the reasoning capabilities"** refers to guiding the LLM to perform multi-step, causal inference. Our two-tier knowledge-enhanced prompt is designed to do this, similar to how Chain-of-Thought (CoT) elicits complex reasoning.
>   - The **insufficiency** of using only triplets is empirically demonstrated in our experiments by the significant performance gap between our method and baselines like K-Paths and KAPING, which use only triplets as input for LLMs. This claim is demonstrated in our ablation studies (e.g., Section 4.3.1 and Table 3), where removing retrieved cases leads to a performance drop, as accurate prediction demands deeper insights into pharmacological mechanisms.
> - **Regarding the sentence: "For example, in Figure 1, the new drug pair..."**
>   - We apologize for the grammatical error and rewrite the sentence: ""For example, as illustrated in Figure 1, the two drugs in the new pair, Fosphenytoin and Diphenhydramine, both bind to the same gene. However, their resulting interaction cannot be directly inferred from this shared property alone without understanding the underlying mechanism."
>   - Our intent was not to imply a literal physical interaction between drug molecules. On the contrary, our method's strength is its ability to reason **beyond** these literal, surface-level physical associations (like two drugs binding to the same gene) provided by a KG.  Such facts are foundational but incomplete. Our method empowers the LLM to use these facts as evidence to **infer the underlying mechanism**—the causal chain of events—that leads to the final, systemic interaction. This is precisely how we overcome the limitations of relying on KG triplets alone, and achieves more accurate prediction.
>
>
>
> We hope these clarifications have adequately addressed your questions. Thank you again for your thoughtful review.

---

> ### Author Response · Authors · 2025-11-25
>
> Dear Reviewer,
>
> We are writing to follow up on your review for our paper. First, we'd like to thank you again for the time you took to provide such detailed and constructive feedback.
>
> We have posted an official comment addressing the concerns you raised, including the usage of LLM, test settings, and controversial sentences.
>
> We would be very grateful if you had a moment to consider our response. We hope it helps address the issues you identified.
> Thank you once again for your valuable engagement with our work.
>
> Sincerely,
>
> Authors

---

### Official Review · Reviewer_DEX8 · 2025-10-31

**Soundness:** 4
**Presentation:** 3
**Contribution:** 4
**Rating:** 6
**Confidence:** 2

**Summary:**

1.This paper proposes CBR-DDI, a framework for DDI reasoning that enhances traditional LLM-based DDI inference by incorporating CBR (Case-Based Reasoning), i.e., historical cases of drug interactions. Subsequent innovations in retrieval mechanisms and knowledge base construction are also designed to support this approach.

 2.For the construction of the DDI knowledge base, unlike traditional methods that rely on knowledge graphs (entity, relation, entity), this paper not only integrates information from authoritative datasets but also incorporates "mechanism insights" and "drug descriptions," which are inferred by the LLM based on knowledge graph information.

  3.Hybrid retrieval mechanism: When performing DDI reasoning, the paper not only uses the structural similarity of drugs to retrieve interaction information from similar drugs as references but also leverages the semantic similarity of drug descriptions. These two similarity scores are weighted and combined to form a final similarity score, which is then used to select reference drugs.

  4.Dual-layer knowledge enhancement: The two layers refer to knowledge from authoritative datasets (internal knowledge) and "mechanism insights" and "drug descriptions" generated by the LLM (internal knowledge). These two types of knowledge are jointly used to prompt the LLM for reasoning.Experimental results demonstrate a 28.7% improvement in accuracy compared to LLM and CBR baselines.

**Strengths:**

1.	The paper is well-structured and easy to follow, allowing readers to quickly grasp the content even without prior specialized knowledge.

2.	The methodology is rigorous, incorporating research and refinements across multiple aspects including knowledge base construction, data retrieval, and knowledge enhancement.

3.	The experiments are comprehensive, featuring not only comparisons with baseline models but also ablative studies evaluating the proposed method itself.

**Weaknesses:**

1.	The paper does not consider the structural information of the drugs themselves, such as the graph structures of molecular or protein-based drugs, relying instead on textual information and interaction relationships.
2.	The study does not incorporate expert evaluation to validate the practical effectiveness of the proposed method.
3.	Both drug descriptions and mechanism insights rely on large language models (LLMs). During inference, the LLM needs to generate three components: drug descriptions, mechanism insights, and reasoning results, resulting in poor inference efficiency. Moreover, could errors in the three outputs generated by the LLM accumulate, leading to progressively amplified errors during the reasoning process?

**Questions:**

N/A

---

> ### Author Response · Authors · 2025-11-21
> **Response to Reviewer DEX8 (Part 1)**
>
> Thanks for your time and valuable feedback. We provide the following response for your concerns:
>
> **W1: About the consideration of the structural information.**
>
> **A1:** Typically, methods for predicting DDI can be categorized into two main types based on their input data: **phenotype-based** and **physiology-based**. Our work falls into the **phenotype-based** category [1, 2], which primarily leverages macroscopic information, such as known DDI networks and external knowledge graphs, for prediction. The **physiology-based** category [3, 4], in contrast, utilizes drug molecular structures to model microscopic interactions. These two approaches are not competing but are fundamentally complementary and orthogonal.
>
> A critical limitation of traditional phenotype-based methods is that they typically only generate final prediction, failing to provide deep, mechanistic explanations for why an interaction occurs. **This is precisely the gap our work aims to fill.** By leveraging the powerful reasoning capabilities of LLMs, our framework uniquely infers plausible interaction mechanisms, providing the kind of insightful, explanatory predictions. In doing so, we enhance the entire category by addressing one of its core limitations, which we believe is a significant contribution.
>
> As we discuss in Appendix A.1, we fully agree that integrating molecular structures is a powerful and complementary direction. In future work, we believe it is highly promising to incorporate molecular structure processing into our framework, which may help with more precise case retrieval and offer deeper pharmacological insights of interaction mechanism.
>
> > [1] Yu, Yue, et al. "SumGNN: multi-typed drug interaction prediction via efficient knowledge graph summarization." Bioinformatics 37.18 (2021): 2988-2995.
> >
> > [2] Zhang, Yongqi, et al. "Emerging drug interaction prediction enabled by a flow-based graph neural network with biomedical network." *Nature Computational Science* 3.12 (2023): 1023-1033.
> >
> > [3] Yang, Ziduo, et al. "Learning size-adaptive molecular substructures for explainable drug–drug interaction prediction by substructure-aware graph neural network." *Chemical science* 13.29 (2022): 8693-8703.
> >
> > [4] Zhong, Yi, et al. "Learning motif-based graphs for drug–drug interaction prediction via local–global self-attention." *Nature Machine Intelligence* 6.9 (2024): 1094-1105.

---

> ### Author Response · Authors · 2025-11-21
> **Response to Reviewer DEX8 (Part 2)**
>
> **W2: About the expert evaluation.**
>
> **A2:** We agree that expert evaluation is the gold standard for clinical validation. While wet-lab experiments are beyond the scope of our current study due to practical constraints, we conduct several case studies to assess the practical effectiveness of our method by comparing its output directly against established findings in professional medical literature. We found high degrees of concordance, as shown in the following examples:
>
> - Example 1: (Oxycodone, Aprepitant)
>   - CBR-DDI Generated Prediction: "Aprepitant is known to inhibit the cytochrome P450 enzyme system, particularly CYP3A4, which is responsible for metabolizing many drugs, including Oxycodone. This inhibition leads to a decreased metabolism of Oxycodone, increasing its serum concentration. Therefore, The metabolism of Oxycodone can be decreased when combined with Aprepitant."
>   - Finding from Published Medical Literature [1]: "…… when using oxycodone metabolized by CYP3A4, it is possible that the effect will be attenuated or enhanced by the concomitant use of drugs that induce or inhibit CYP3A4. Aprepitant, an antiemetic drug used in many patients receiving anticancer drugs, is known as a moderate competitive inhibitor of CYP3A4. We experienced a case of respiratory depression caused by opioids, which was suspected to be caused by a drug interaction with antiemetics especially aprepitant."
> - Example 2: (Fluticasone furoate, Oxandrolone)
>   - CBR-DDI Generated Prediction: "The related facts provided suggest that Fluticasone furoate resembles other corticosteroids, which have been shown to increase the fluid-retaining activities of Oxandrolone. Since Fluticasone furoate is structurally similar to these corticosteroids, it is likely that it will have a similar effect on Oxandrolone. Therefore, Fluticasone furoate may increase the fluid retaining activities of Oxandrolone."
>   - Finding from Published Medical Literature [2]:"In addition, the use of adrenal cortical steroids or adrenocorticotropic hormone (ACTH) with oxandrolone may also exacerbate edema." (Fluticasone furoate is a corticosteroid, and "exacerbating edema" is a direct clinical consequence of increased fluid-retaining activity.)
> - Example 3: (Rifabutin, Felodipine)
>   - CBR-DDI Generated Prediction: "Rifamycins can induce the cytochrome P450 enzyme system, particularly CYP3A4, which is responsible for metabolizing many drugs, including Felodipine. This induction leads to an increased metabolism of Felodipine, reducing its serum concentration and potentially its therapeutic effect. Therefore, The metabolism of Felodipine can be increased when combined with Rifabutin."
>   - Finding from Published Medical Literature [3] in TABLE 12. CLINICALLY SIGNIFICANT DRUG–DRUG INTERACTIONS INVOLVING THE RIFAMYCINS: "a similar interaction is also predicted for cardiovascular agent felodipine and nisoldipine. ……Clinical monitoring recommended; may require change to an alternate cardiovascular agent."
>
> These cases demonstrate that our method can produce outputs that are not only accurate but also consistent with expert clinical knowledge, highlighting its potential for practical application.
>
>
> > [1] Hamada, Hiroshi, Suzuki, Eisuke, Endo, Mitsufumi, Mihara, Yukiko, Iketani, Sayaka, Ishida, Miki, Miyasato, Akime, AND Miyazaki, Kanako. "Opioid-induced respiratory depression suspected of drug interaction in a prostate cancer patient: a case report" *Annals of Palliative Medicine*, Volume 13 Number 2 (26 March 2024)
> >
> > [2] Kopel J, Sorensen G, Griswold J. A Reappraisal of Oxandrolone in Burn Management. J Pharm Technol. 2022 Aug;38(4):232-238. doi: 10.1177/87551225221091115. Epub 2022 May 3. PMID: 35832568; PMCID: PMC9272491
> >
> > [3] Blumberg, Henry M., et al. "American thoracic society/centers for disease control and prevention/infectious diseases society of America: treatment of tuberculosis." *American journal of respiratory and critical care medicine* 167.4 (2003): 603.

---

> ### Author Response · Authors · 2025-11-21
> **Response to Reviewer DEX8 (Part 3)**
>
> **W3: About inference efficiency and error accumulation.**
>
> **A3:** Thank you for raising these important practical considerations. We have designed our framework to be both efficient and robust.
>
> - **On Inference Efficiency:**
>   The **mechanistic insights and the final prediction result are generated sequentially within a single LLM call**. Therefore, our end-to-end pipeline requires only **two LLM calls** per prediction: one for the initial drug descriptions and one for the final reasoning step. To further enhance efficiency, it can be optimized by using a smaller, faster model (e.g., an 8B model) for description generation, reserving a more powerful model for the complex reasoning task.
> - **On Error Accumulation:**
>   1. **High Accuracy of the first Step:** We first validat the drug description generation. We randomly sampled 50 drug descriptions generated by Llama3.1-8B-Instruct, and prompt a powerful model (DeepSeek-R1) to compare our generated descriptions against the ground-truth descriptions from the DrugBank database. **The evaluation showed a 98% accuracy rate**. This high fidelity ensures the reasoning process starts from a factually sound foundation.
>   2. **Linked Generation in the Second Step:** The subsequent components—the mechanistic insight and the final answer—are generated together in one continuous process. The final answer is directly derived from the insight generated in the same step. Because these two outputs are intrinsically linked, there is no opportunity for errors to be amplified between them.
>
>
>
>
> We hope these clarifications have adequately addressed your questions. Thank you again for your thoughtful review.

---

### Official Review · Reviewer_LiBx · 2025-10-31

**Soundness:** 3
**Presentation:** 3
**Contribution:** 1
**Rating:** 2
**Confidence:** 4

**Summary:**

This paper tackles the problem of drug-drug interaction prediction. Based on a given biological knowledge graph, the method first builds a knowledge repository of previously known drug-drug interaction cases (this process is made efficient with representative sampling). At inference time, the system retrieves relevant historical cases using semantic and structural similarity. The retrieved cases are then incorporated into a prompt to a LLM that performs the final prediction. The authors evaluate their method on the DrugBank and TWOSIDES datasets and show that it improves over both graph based methods and naive LLM methods by a sizeable margin.

**Strengths:**

- This work proposes a very effective solution for an important and impactful problem (drug drug interaction prediction)
- The paper is clearly written and easy to follow
- The authors run extensive ablations to study the impact of each of the components.

**Weaknesses:**

- The major weakness of this work is its relevance for a machine learning conference like ICLR. The approach is elegant and outperforms baselines on that particular problem. However, there is no substantial machine learning contributions. That said the paper is really creative way to solve the drug-drug interaction problem and might better shine in a venue more focused on that particular problem.

- The architecture of CBR-DDI is highly tailored to the drug drug interaction problem and it's not clear whether it could be applied easily in other contexts.

**Questions:**

- Do the authors think that the approach presented here could benefit other application areas ?

---

> ### Author Response · Authors · 2025-11-21
> **Response to Reviewer LiBx (Part 1)**
>
> Thank you for your valuable time and feedback on our manuscript. We would like to take this opportunity to address your concerns regarding the paper's relevance to ICLR and the generalizability of our framework.
>
> **W1： About the relevance to ICLR.**
>
> **A1:** We respectfully wish to clarify our work's positioning and contributions, which we believe are highly relevant to the ICLR community from both a machine learning and an application perspective.
>
> **From the perspective of machine learning contribution:**  First and foremost, we believe that **advancing machine learning by tackling complex, high-impact interdisciplinary challenges is a significant contribution in its own right.** Such applications not only demonstrate the utility of our methods but also push the boundaries of what is possible, revealing new research problems and requirements for the ML community. Beyond this, our work introduces several specific, novel components that advance the field of deep learning, particularly for complex reasoning tasks: Our primary contributions are:
>
> - **A New Framework for LLM Reasoning via Case-Based Reasoning (CBR):** We propose a new paradigm that formally integrates the classic AI paradigm of CBR with modern LLMs. This moves beyond simple few-shot prompting by creating a structured repository of cases with distilled, transferable mechanistic patterns, providing a principled way to enhance the analogical reasoning capabilities of LLMs.
> - **A Hybrid LLM-GNN Architecture for Neuro-Symbolic Retrieval:** We introduce a symbiotic retrieval mechanism where an LLM and a GNN collaborate. This neuro-symbolic approach, which balances semantic and structural similarity, is a sophisticated architectural contribution to the Retrieval-Augmented Generation (RAG) landscape.
> - **A Two-Tier Knowledge-Enhanced Prompting Strategy:** We design a principled prompting method that guides the LLM by explicitly separating external factual knowledge (from a KG) and internal regularity knowledge (from historical cases). This structured guidance enhances reasoning accuracy and interpretability.
>
> **From the perspective of the application domain:** Demonstrating these ML advancements on challenging real-world problems is well-aligned with the scope of ICLR. The conference has a dedicated track for **"applications to physical sciences (physics, chemistry, biology, etc.)"** and has shown a strong and growing interest in this area.
>
> - For instance, the ICLR 2025 program for this track includes 178 papers, with **9 papers specifically focusing on drug-related problems** (including 2 selected for Oral presentations). Examples include:
>
> > Zhang, Chenbin, et al. "Rethinking the generalization of drug target affinity prediction algorithms via similarity aware evaluation." *The Thirteenth International Conference on Learning Representations*. (Oral)
>
> > Adams, Keir, et al. "ShEPhERD: Diffusing shape, electrostatics, and pharmacophores for bioisosteric drug design." *The Thirteenth International Conference on Learning Representations*. (Oral)
>
> > Li, Ruifeng, et al. "UniMatch: Universal Matching from Atom to Task for Few-Shot Drug Discovery." *The Thirteenth International Conference on Learning Representations*. (Poster)
>
> This trend is not limited to ICLR; other premier machine learning conferences also consistently feature high-impact research in this domain. For example:
>
> > Ma, Tengfei, et al. "Self-supervised Blending Structural Context of Visual Molecules for Robust Drug Interaction Prediction." (NeurIPS 2025)
>
> > Bang, Dongmin, et al. "BounDr. E: Predicting Drug-likeness via Biomedical Knowledge Alignment and EM-like One-Class Boundary Optimization." (ICML 2025)
>
> > Du, Haotong, et al. "Customized subgraph selection and encoding for drug-drug interaction prediction."(NeurIPS 2024)
>
> Given our contributions to LLM reasoning and hybrid architectures, and the clear precedent for this application domain at the conference, we are confident our work is an excellent fit for the ICLR community.

---

> ### Author Response · Authors · 2025-11-21
> **Response to Reviewer LiBx (Part 2）**
>
> **W2 & Q1：About how the approach benefits other application areas.**
>
> **A2:** Thank you for this insightful question. It is true that our CBR-DDI framework was carefully designed to address the unique challenges of the drug-drug interaction (DDI) task. The DDI problem is defined by a key set of characteristics: a successful prediction for any given drug pair requires a deep understanding of two complementary sources of information—the rich **semantic context** surrounding the pair (including their functional descriptions and potential interaction mechanisms) and their complex **structural relationships** within a biomedical KG. Furthermore, effective solutions in this domain are often derived by **adapting known solutions from analogous past cases**.
>
> Many other complex domains share these same foundational characteristics. For instance, challenging problems in areas such as **Financial Fraud Detection**, **Legal Case Precedent Retrieval**, and **Personalized Recommendation Systems** also require reasoning over both unstructured semantic data (like text descriptions or user reviews) and structured relational data (like transaction networks or user-item interaction graphs). Moreover, in these fields, employing reasoning by analogy from prior cases facilitates effective problem-solving. Therefore, our framework offers an effective and generalizable template for enhancing reasoning across this wide range of important applications.
>
>
>
> We hope this response clarifies the machine learning novelty and broad potential of our work. Thank you once again for your time and feedback.

---

### Meta-Review · Area_Chair_JUFQ · 2026-01-02

**Summary:**

The paper proposes CBR-DDI, a framework to enhance LLM reasoning for DDI prediction using Case-Based Reasoning. The method constructs a knowledge repository by using an LLM to distill mechanistic insights and a Graph Neural Network to extract drug associations from a biomedical Knowledge Graph. It utilizes a hybrid retrieval mechanism (combining semantic and structural similarity) to find historical cases and employs a two-tier knowledge-enhanced prompting strategy to predict interactions. The authors report state-of-the-art performance improvements over baselines based on DrugBank/TWOSIDES datasets.

However, the paper suffers from significant limitations regarding its technical novelty and practical applicability. From an AI/ML perspective, the work combines existing techniques without introducing substantial ML algorithmic contributions. From an application perspective, the problem formulation is viewed as overly simplistic because it treats DDI prediction as a static classification task while ignoring essential real-world pharmacological factors like dosage, timing, and route of administration. Finally, the approach ignores molecular structure information, which is a vital data source for DDI prediction, and lacks rigorous experimental or clinical validation to confirm the utility of the LLM-generated mechanistic insights.

**Reviewer Concerns:**

Addressed Concerns:

- The authors clarified the specific roles of the LLM and GNN, and the definition of "new drugs" within their experimental setup (drugs present in the KG but without DDI labels in the training set).

- In response to Reviewers 6rWq and DEX8 regarding the hallucination risks of LLM-generated mechanistic insights, the authors conducted a consistency check using a stronger model (DeepSeek-R1) on a small sample (50 descriptions), reporting 98% consistency with DrugBank.




Outstanding Concerns:

- Reviewer LiBx raised strong concerns that the paper lacks substantial machine learning contributions and is "not relevant for a machine learning conference like ICLR". While the authors argued that ICLR has an "AI for Science" track, the core concern regarding the depth of the contribution remains. As the AC, I judge that while the topic of drug discovery is highly relevant, the paper does not make significant technical contributions to ML algorithms, nor does it demonstrate a transformative impact on DDI prediction that would justify acceptance purely on application grounds.

- The proposed formulation is overly simplistic for real-world DDI discovery. It treats DDI as a static binary/multi-class prediction based on text and graph neighbors, ignoring critical pharmacological factors such as timing, route of administration, and dosages. The paper also lacks a discussion on how these predictions could be validated experimentally or clinically, limiting its "AI for Science" value.

- Reviewer kDBh questioned the model's ability to handle truly new drugs that do not appear in the Knowledge Graph. The authors argued this is a "misunderstanding" and that the standard setting assumes KG presence. However, I agree with the reviewer: for a method claiming to solve "cold-start" or new drug problems, experiments on de novo compounds (absent from the KG) are necessary to prove the method is not just retrieving existing property data.

- The method relies on phenotype/text information and ignores molecular structure, which the authors admitted is a limitation left for future work in the reply to Reviewer DEX8.

**Reviewer Scores:**

I don't think reviewers will change their scores.

---

### Decision · Program_Chairs · 2026-01-26

Reject